# Neural Unbalanced Optimal Transport via Cycle-Consistent Semi-Couplings

## Abstract

Comparing *unpaired* samples of a distribution or population taken at different points in time is a fundamental task in many application domains where measuring populations is destructive and cannot be done repeatedly on the same sample, such as in single-cell biology. Optimal transport (OT) can solve this challenge by learning an optimal coupling of samples across distributions from unpaired data. However, the usual formulation of OT assumes conservation of mass, which is violated in *unbalanced* scenarios in which the population size changes (e.g., cell proliferation or death) between measurements. In this work, we introduce NUBOT, a *neural unbalanced OT* formulation that relies on the formalism of *semi-couplings* to account for creation and destruction of mass. To estimate such semi-couplings and generalize out-of-sample, we derive an efficient parameterization based on neural optimal transport maps and propose a novel algorithmic scheme through a cycle-consistent training procedure. We apply our method to the challenging task of forecasting heterogeneous responses of multiple cancer cell lines to various drugs, where we observe that by accurately modeling cell proliferation and death, our method yields notable improvements over previous neural optimal transport methods.

## 1 Introduction

Modeling change is at the core of various problems in the natural sciences, from dynamical processes driven by natural forces to population trends induced by interventions. In all these cases, the gold standard is to track particles or individuals across time, which allows for immediate estimation of individual (or aggregate) effects. But maintaining these pairwise correspondences across interventions or time is not always possible, for example, when the same sample cannot be measured more than once. This is typical in biomedical sciences, where the process of measuring is often altering or destructive. For example, single-cell biology profiling methods destroy the cells and thus cannot be used repeatedly on the same cell. In these situations, one must rely on comparing *different* replicas of a population and, absent a natural identification of elements across the populations, infer these correspondences from data in order to model evolution or intervention effects.

The problem of inferring correspondences across unpaired samples in biology has been traditionally tackled by relying on average and aggregate perturbation responses (Green & Pelkmans, 2016; Zhan et al., 2019; Sheldon et al., 2007) or by applying mechanistic or linear models (Yuan et al., 2021; Dixit et al., 2016) in, potentially, a learned latent space (Lotfollahi et al., 2019). Cellular responses to treatments are, however, highly complex and heterogeneous. To effectively predict the drug response of a patient during treatment and capture such cellular heterogeneity, it is necessary to learn nonlinear maps describing such perturbation responses on the level of single cells. Assuming perturbations incrementally alter molecular profiles of cells, such as gene expression or signaling activities, recent approaches have utilized optimal transport to predict changes and alignments (Schiebinger et al., 2019; Bunne et al., 2022a; Tong et al., 2020). By returning a coupling between control and perturbed cell states, which overall minimizes the cost of matching, optimal transport can solve that puzzle and reconstruct these incremental changes in cell states over time.

Despite the advantages mentioned above, the classic formulation of OT is ill-suited to model processes where the population changes in *size*, e.g., where elements might be created or destroyed over time. This is the case, for example, in single-cell biology, where interventions of interest typically promote proliferation of certain cells and death of others. Such scenarios violate the assumption of conservation

Figure 1: **a.** A semi-coupling pair ($\gamma_1$, $\gamma_2$) consists of two couplings that together solve the *unbalanced* OT problem. Intuitively, $\gamma_1$ describes where mass goes as it leaves from $\mu$, and $\gamma_2$ where it comes from as it arrives in $\nu$. **b.** NUBOT parameterizes the semi-couplings ($\gamma_1$, $\gamma_2$) as the composition of reweighting functions $\eta$ and $\zeta$ and the dual potentials $f$ and $g$ between the then *balanced* problem.

of mass that the classic OT problem relies upon. Relaxing this assumption yields a generalized formulation, known as the *unbalanced* OT (UBOT) problem, for which recent work has studied its properties (Liero et al., 2018; Chizat et al., 2018a), numerical solution (Chapel et al., 2021), and has applied it successfully to problems in single cell biology (Yang & Uhler, 2019). Yet, these methods typically scale poorly with sample size, are prone to unstable solutions, or make limiting assumptions, e.g., only allowing for destruction but not creation of mass.

In this work, we address these shortcomings by introducing a novel formulation of the unbalanced OT problem that relies on the formalism of semi-couplings introduced by Chizat et al. (2018b), while still obtaining an explicit transport map that models the transformation between distributions. The advantage of the latter is that it allows mapping new out-of-sample points, and it provides an interpretable characterization of the underlying change in distribution. Since the unbalanced OT problem does not directly admit a Monge (i.e., mapping-based) formulation, we propose to *learn* to jointly 're-balance' the two distributions, thereby allowing us to estimate a map between their rescaled versions. To do so, we leverage prior work (Makkuva et al., 2020; Korotin et al., 2021) that learns the transport map as the gradient of a convex dual potential (Brenier, 1987) parameterized via an input convex neural network (Amos et al., 2017). In addition, we derive a simple update rule to learn the rescaling functions. Put together, these components yield a reversible, parameterized, and computationally feasible implementation of the semi-coupling unbalanced OT formulation (Fig. 1).

In short, the main **contributions** of this work are: (i) A novel formulation of the unbalanced optimal transport problem that weaves together the theoretical foundations of semi-couplings with the practical advantage of transport maps; (ii) A general, scalable, and efficient algorithmic implementation for this formulation based on dual potentials parameterized via convex neural network architectures; and (iii) An empirical validation on the challenging task of predicting perturbation responses of single cells to multiple cancer drugs, where our method successfully predicts cell proliferation and death, in addition to faithfully modeling the perturbation responses on the level of single cells.

## 2 BACKGROUND

### 2.1 OPTIMAL TRANSPORT

For two probability measures $\mu, \nu$ in $\mathcal{P}(\mathcal{X})$ with $\mathcal{X} = \mathbb{R}^d$ and a real-valued continuous cost function $c \in \mathcal{C}(\mathcal{X}^2)$, the optimal transport problem (Kantorovich, 1942) is defined as

$$\text{OT}(\mu, \nu) := \inf_{\gamma \in \Gamma(\mu, \nu)} \int_{\mathcal{X}^2} c(x, y) \gamma(dx, dy), \tag{1}$$

where $\Gamma(\mu, \nu) = \{\gamma \in \mathcal{M}_+(\mathcal{X}^2), \text{ s.t. } (\text{Proj}_1)_\sharp \gamma = \mu, (\text{Proj}_2)_\sharp \gamma = \nu\}$ is the set of couplings in the cone of nonnegative Radon measures $\mathcal{M}_+(\mathcal{X}^2)$ with respective marginals $\mu, \nu$. When instantiated on finite discrete measures, such as $\mu = \sum_{i=1}^n u_i \delta_{\mathbf{x}_i}$ and $\nu = \sum_{j=1}^m v_j \delta_{\mathbf{y}_j}$, with $\mathbf{u} \in \Sigma_n, \mathbf{v} \in \Sigma_m$ this problem translates to a linear program, which can be regularized using an entropy term (Peyré & Cuturi, 2019). For $\varepsilon \geq 0$, set

$$\text{OT}_\varepsilon(\mu, \nu) := \min_{\mathbf{P} \in U(\mathbf{u}, \mathbf{v})} \langle \mathbf{P}, [c(\mathbf{x}_i, \mathbf{y}_j)]_{ij} \rangle - \varepsilon H(\mathbf{P}), \tag{2}$$

where $H(\mathbf{P}) := -\sum_{ij} \mathbf{P}_{ij}(\log \mathbf{P}_{ij} - 1)$ and the polytope $U(\mathbf{u}, \mathbf{v})$ is the set of matrices $\{\mathbf{P} \in \mathbb{R}_+^{n \times m}, \mathbf{P}\mathbf{1}_m = \mathbf{u}, \mathbf{P}^\top \mathbf{1}_n = \mathbf{v}\}$. For clarity, we will sometimes write $\mathrm{OT}_\varepsilon(\mathbf{u}, \mathbf{v}, \{\mathbf{x}_i\}, \{\mathbf{y}_j\})$. Notice that the definition above reduces to (1) when $\varepsilon = 0$. Setting $\varepsilon > 0$ yields a faster and differentiable proxy to approximate OT and allows fast numerical approximation via the Sinkhorn algorithm (Cuturi, 2013), but introduces a bias, since in general $\mathrm{OT}_\varepsilon(\mu, \mu) \neq 0$.

**Neural optimal transport.** To parameterize (1) and allow to predict how a measure evolves from $\mu$ to $\nu$, we introduce an alternative formulation known as the Monge problem (1781) given by

$$\mathrm{OT}(\mu, \nu) = \inf_{T : T_\sharp \mu = \nu} \int_{\mathcal{X}} c(x, T(x)) d\mu(x), \tag{3}$$

with pushforward operator $\sharp$ and the optimal solution $T^*$ known as the Monge map between $\mu$ and $\nu$. When cost $c$ is the quadratic Euclidean distance, i.e., $c = \|\cdot\|_2^2$, Brenier's theorem (1987) states that this Monge map is necessarily the gradient $\nabla\psi$ of a convex potential $\psi : \mathcal{X} \mapsto \mathbb{R}$ such that $\nabla\psi_\sharp \mu = \nu$, i.e., $T^*(x) = \nabla\psi(x)$. This connection has far-reaching impact and is a central component of recent neural optimal transport solvers (Makkuva et al., 2020; Bunne et al., 2022c; Alvarez-Melis et al., 2022; Korotin et al., 2020; Bunne et al., 2022b; Fan et al., 2021b). Instead of (indirectly) learning the Monge map $T$ (Yang & Uhler, 2019; Fan et al., 2021a), it is sufficient to restrict the computational effort to learning a *good* convex potential $\nabla_\theta$, parameterized via input convex neural networks (ICNN) (Amos et al., 2017), s.t. $\nabla_\theta\psi_\sharp \mu = \nu$. Alternatively, parameterizations of such maps can be carried out via the dual formulation of (1) (Santambrogio, 2015, Proposition 1.11, Theorem 1.39), i.e.,

$$\mathrm{OT}(\mu, \nu) := \sup_{\substack{f, g \in \mathcal{C}(\mathcal{X}) \\ f \oplus g \leq c}} \int f d\mu + \int g d\nu, \tag{4}$$

where the dual potentials $f, g$ are continuous functions from $\mathcal{X}$ to $\mathbb{R}$, and $f \oplus g \mapsto f(x) + g(x)$. Based on Brenier (1987), Makkuva et al. (2020) derive an approximate min-max optimization scheme parameterizing the duals $f, g$ via two convex functions. The objective thereby reads

$$\mathrm{OT}(\mu, \nu) = \sup_{f \text{ convex}} \inf_{g \text{ convex}} \underbrace{\frac{1}{2}\mathbb{E}\left[\|x\|_2^2 + \|y\|_2^2\right]}_{\mathcal{C}_{\mu,\nu}} - \underbrace{\mathbb{E}_\mu[f(x)] - \mathbb{E}_\nu[\langle y, \nabla g(y)\rangle - f(\nabla g(y))]}_{\mathcal{V}_{\mu,\nu}(f,g)}. \tag{5}$$

When parameterizing $f$ and $g$ via a pair of ICNNs with parameters $\theta_f$ and $\theta_g$, this neural OT scheme then allows to predict $\nu$ or $\mu$ via $\nabla g_{\theta_g \sharp}\mu$ or $\nabla f_{\theta_f \sharp}\nu$, respectively. We further discuss neural primal (Fan et al., 2021a; Yang & Uhler, 2019) and dual approaches (Makkuva et al., 2020; Korotin et al., 2020; Bunne et al., 2021) in §D.2.

## 2.2 UNBALANCED OPTIMAL TRANSPORT

A major constraint of problem (1) is its restriction to a pair of probability distributions $\mu$ and $\nu$ of equal mass. Unbalanced optimal transport (Benamou, 2003; Liero et al., 2018; Chizat et al., 2018b) lifts this requirement and allows a comparison between unnormalized measures, i.e., via

$$\inf_{\gamma \in \mathcal{M}_+(\mathcal{X}^2)} \int_{\mathcal{X}^2} c(x, y)\gamma(dx, dy) + \tau_1 \mathcal{D}_{f_1}((\mathrm{Proj}_1)_\sharp \gamma \mid \mu) + \tau_2 \mathcal{D}_{f_2}((\mathrm{Proj}_2)_\sharp \gamma \mid \nu), \tag{6}$$

with $f$-divergences $\mathcal{D}_{f_1}$ and $\mathcal{D}_{f_2}$ induced by $f_1, f_2$, and parameters $(\tau_1, \tau_2)$ controlling how much mass variations are penalized as opposed to transportation of the mass. When introducing an entropy regularization as in (2), the unbalanced OT problem between discrete measures $\mathbf{u}$ and $\mathbf{v}$, i.e.,

$$\mathrm{UBOT}(\mathbf{u}, \mathbf{v}) := \min_{\Gamma \in \mathbb{R}_+^{n \times m}} \langle \Gamma, [c(x_i, y_j)]_{ij} \rangle + \tau_1 \mathcal{D}_{f_1}(\Gamma \mathbb{1}_n \mid \mathbf{u}) + \tau_2 \mathcal{D}_{f_2}(\Gamma^\top \mathbb{1}_m \mid \mathbf{v}) - \varepsilon H(\Gamma), \tag{7}$$

can be efficiently solved via generalizations of the Sinkhorn algorithm (Chizat et al., 2018a; Cuturi, 2013; Benamou et al., 2015) . We describe alternative formulations of the unbalanced OT problem in detail, review recent applications, and provide a broader literature review in the Appendix (§A.1).

## 3 A NEURAL UNBALANCED OPTIMAL TRANSPORT MODEL

The method we propose weaves together a rigorous formulation of the unbalanced optimal transport problem based on semi-couplings (introduced below) with a practical and scalable OT mapping estimation method based on input convex neural network parameterization of the dual OT problem.

**Semi-coupling formulation.** Chizat et al. (2018b) introduced a class of distances that generalize optimal transport for the unbalanced setting. They introduce equivalent dynamic and static formulations of the problem, the latter of which relies on *semi-couplings* to allow for variations of mass. A formulation closely related (by a change of variables) to the notion of semi-couplings was independently introduced by Liero et al. (2016; 2018). These are generalizations of couplings whereby only one of the projections coincides with a prescribed measure. Formally, the set of semi-couplings between measures $\mu$ and $\nu$ is defined as

$$\Gamma_{\shortparallel}(\mu, \nu) \stackrel{\text{def.}}{=} \left\{ (\gamma_0, \gamma_1) \in \left( \mathcal{M}_+ \left( \mathcal{X}^2 \right) \right)^2 : (\mathrm{Proj}_1)_\sharp \gamma_0 = \mu, (\mathrm{Proj}_2)_\sharp \gamma_1 = \nu \right\}. \tag{8}$$

With this, the unbalanced Kantorovich OT problem can be written as $C_k(\mu, \nu) = \inf_{(\gamma_0, \gamma_1) \in \Gamma(\mu, \nu)} \int c(x, \frac{\gamma_0}{\gamma}, y, \frac{\gamma_1}{\gamma}) \, \mathrm{d}\gamma(x, y)$, where $\gamma$ is any joint measure for which $\gamma_0, \gamma_1 \ll \gamma$.

Although this formulation lends itself to formal theoretical treatment, it has at least two limitations. First, it does not explicitly model a mapping between measures. Indeed, no analogue of the celebrated Brenier's theorem is known for this setting. Second, deriving a computational implementation of this problem is challenging by the very nature of the semi-couplings: being undetermined along one marginal makes it hard to model the space in (8).

**Rebalancing with proxy measures.** To turn the semi-coupling formulation of unbalanced OT into a computationally feasible method, we propose to conceptually break the problem into balanced and unbalanced subproblems, each tackling a different aspect of the difference between measures: feature transformation and mass rescaling. These in turn imply a decomposition of the semi-couplings of (8), as we will show later. Specifically, we seek proxy measures $\tilde{\mu}$ and $\tilde{\nu}$ with equal mass (i.e., $\mu(\mathcal{X}) = \tilde{\nu}(\mathcal{X})$) across which to solve a *balanced* OT problem through a Monge/Brenier formulation. To decouple measure scaling from feature transformation, we propose to choose $\tilde{\mu}$ and $\tilde{\nu}$ simply as rescaled versions of $\mu$ and $\nu$. Thus, formally, we seek $\tilde{\mu}, \tilde{\nu} \in \mathcal{M}_+(\mathcal{X})$ and $T, S : \mathcal{X} \to \mathcal{X}$ such that

$$\tilde{\mu} = \eta \cdot \mu, \quad \tilde{\nu} = \zeta \cdot \nu, \quad T_\sharp \tilde{\mu} = \tilde{\nu}, \quad S_\sharp \tilde{\nu} = \tilde{\mu}, \tag{9}$$

where $\eta, \zeta : \mathcal{X} \to \mathbb{R}^+$ are scalar fields, $\eta \cdot \mu$ denotes the measure with density $\eta(x) \, \mathrm{d}\mu(x)$ (analogously for $\zeta \cdot \nu$), and $T, S$ are a pair of forward/backward optimal transport maps between $\tilde{\mu}$ and $\tilde{\nu}$ (Fig. 1b).

Devising an optimization scheme to find all relevant components in (9) is challenging. In particular, it involves solving an OT problem whose marginals are not fixed, but that will change as the reweighting functionals $\eta, \zeta$ are updated. We propose an alternating minimization approach, whereby we alternative solve for $\eta, \zeta$ (through an approximate scaling update) and $T, S$ (through gradient updates on ICNN convex potentials, as described in Section 2.1).

**Updating rescaling functions.** Given current estimates of $\eta$ and $T$, we consider the UBOT problem (6) between $T_\sharp(\eta \cdot \mu) = T_\sharp \tilde{\mu}$ and $\nu$. Although in general these two measures will not be balanced (hence why we need to use UBOT instead of usual OT), our goal is to eventually achieve this. To formalize this, let us use the shorthand notation $\gamma_{\mathrm{UB}}^*(\alpha, \beta) := \mathrm{argmin}_\gamma \mathrm{UBOT}(\gamma; \alpha, \beta)$, where UBOT is defined in (7). For a fixed $T$, our goal is to find $\eta$ such that $(\mathrm{Proj}_1)_\sharp [\gamma_{\mathrm{UB}}^*(T_\sharp(\eta \cdot \mu), \nu)] = T_\sharp(\eta \cdot \mu)$, i.e., to rescale $\mu$ so that the unbalanced solution would in fact be 'balanced' along that marginal. For the discrete setting (finite samples), this corresponds to finding a vector $\mathbf{e} \in \mathbb{R}^n$ satisfying:

$$\sum_{j=1}^m [\mathbf{\Gamma}]_{ij} = \mathbf{e} \odot \mathbf{u}, \quad \text{where } \mathbf{\Gamma} = \mathrm{argmin}\, \mathrm{UBOT}(\mathbf{e} \odot \mathbf{u}, T(\mathbf{x}_i), \mathbf{v}, \mathbf{y}_j). \tag{10}$$

For a fixed $T$, the vector $\mathbf{e}^*$ satisfying this system can be found via a fixed-point iteration. In practice, we approximate it instead with a single-step update using the solution to the unscaled problem:

$$\mathbf{\Gamma} \leftarrow \mathrm{argmin}\, \mathrm{UBOT}(\mathbf{u}, T(\mathbf{x}_i), \mathbf{v}, \mathbf{y}_j); \qquad \mathbf{e} \leftarrow \mathbf{\Gamma}\mathbb{1} \oslash \mathbf{u} \tag{11}$$

which empirically provides a good approximation on the optimal $\mathbf{e}^*$ but is significantly more efficient. Apart from requiring a single update, whenever $\mathbf{u}$ and $\mathbf{v}$ are uniform (as in most applications where the samples are assumed to be drawn i.i.d.) solving this problem between unscaled histograms will be faster and more stable than solving its scaled (and therefore likely non-uniform) counterpart in (10).

Analogously, for a given $S$, we choose $\zeta$ that ensures $(\mathrm{Proj}_2)_\sharp [\gamma_{\mathrm{UB}}^*(S_\sharp(\zeta \cdot \nu), \mu)] = S_\sharp(\zeta \cdot \nu)$. For empirical measures, this yields the update:

$$\mathbf{\Gamma} \leftarrow \mathrm{argmin}\, \mathrm{UBOT}(\mathbf{v}, S(\mathbf{y}_j), \mathbf{u}, \mathbf{x}_i); \qquad \mathbf{z} \leftarrow \mathbf{\Gamma}\mathbb{1} \oslash \mathbf{v}; \tag{12}$$

---

**Algorithm 1** Neural Unbalanced Optimal Transport (NUBOT)

---

**Input:** $f, g$: ICNNs, initialized s.t. $\nabla g(x) \approx x$ and $\nabla f(y) \approx y$ ; $\eta, \zeta$: NNs

1 **for** t in epochs **do**
2     Sample batch $\{x_i\}_{i=1}^n \sim \mu$ and $\{y_j\}_{j=1}^m \sim \nu$
3     $\hat{y} \leftarrow \nabla g(x)$
4     $\hat{x} \leftarrow \nabla f(y)$
    /* Marginal fitting update Eq. (11)                                               */
5     $\Gamma_1 \leftarrow \texttt{unbalanced.sinkhorn}(\hat{y}, \frac{1}{n}\mathbb{1}_n, y, \frac{1}{m}\mathbb{1}_m)$
6     $e_i \leftarrow \frac{\sum_j \Gamma_{ij}}{\sum_{ij} \Gamma_{ij}} \cdot n$
    /* Marginal fitting update Eq. (12)                                               */
7     $\Gamma_2 \leftarrow \texttt{unbalanced.sinkhorn}(\hat{x}, \frac{1}{m}\mathbb{1}_m, x, \frac{1}{n}\mathbb{1}_n)$
8     $z_i \leftarrow \frac{\sum_j \Gamma_{ij}}{\sum_{ij} \Gamma_{ij}} \cdot m$
9     $J(\theta_g, \theta_f) = \frac{1}{n}\sum_{i=1}^n e_i \left[ f(\nabla g(x_i)) - \langle x_i, \nabla g(x_i)\rangle \right] - \frac{1}{m}\sum_{j=1}^m z_j f(y_j)$
10     $L_\eta(\theta_\eta) = \texttt{MSE}(\mathbf{e}, \eta(x))$
11     $L_\zeta(\theta_\zeta) = \texttt{MSE}(\mathbf{z}, \zeta(y))$
12     Update $\theta_g$ to minimize $J$, $\theta_\eta$ to minimize $L_\eta$, $\theta_\zeta$ to minimize $L_\zeta$, and $\theta_f$ to maximize $J$

---

In order to be able to predict mass changes for new samples, we will use the discrete $\mathbf{e}, \mathbf{z}$ to fit continuous versions of $\eta, \zeta$ via functions parameterized as neural networks trained to achieve $\eta(\mathbf{x}_i) \approx e_i \ \forall i \in \{1, \ldots, n\}$ and $\zeta(\mathbf{y}_j) \approx z_j \ \forall j \in \{1, \ldots, m\}$ through a mean squared error loss.

**Updating mappings.** Since $\eta, \zeta$ are tasked with modeling all mass rescalings, for fixed rescaled measures $\tilde{\mu}$ and $\tilde{\nu}$ we can model the transformation between them with a usual (balanced) OT formulation, whereby we seek $T$ and $S$ to be a pair of optimal (deterministic) OT maps between them. In particular, we use the formulation of (Makkuva et al., 2020) to fit them. That is, $T = \nabla g$ and $S = \nabla f$ for convex potentials $f$ and $g$, parameterized as ICNNs with parameters $\theta_f$ and $\theta_g$. The corresponding objective for these two potentials is:

$$\mathcal{L}(f, g) = \mathop{\mathbb{E}}_{x \sim \tilde{\mu}} \left[ f(\nabla g(x)) - \langle x, \nabla g(x)\rangle \right] - \mathop{\mathbb{E}}_{y \sim \tilde{\nu}} \left[ f(y) \right]$$
$$= \int_\mathcal{X} \left[ f(\nabla g(x)) - \langle x, \nabla g(x)\rangle \right] \eta(x)\, \mathrm{d}\mu(x) - \int f(y)\zeta(y)\, \mathrm{d}\nu(y).$$

In the finite sample setting, this objective becomes:

$$\mathcal{L}(f, g) = \frac{1}{n}\sum_{i=1}^n e_i \left[ f(\nabla g(\mathbf{x}_i)) - \langle \mathbf{x}_i, \nabla g(\mathbf{x}_i)\rangle \right] - \sum_{j=1}^m z_j f(\mathbf{y}_j). \tag{13}$$

The optimization procedure is summarized in Algorithm 1.

**Transforming new samples.** After learning $f, g, \eta, \zeta$, we can use these functions to transform (map and rescale) new samples, i.e., beyond those used for optimization. For a given source datapoint $\mathbf{x}$ with mass $u$, we transform it as $(\mathbf{x}, u) \mapsto (\nabla g(\mathbf{x}), \eta(\mathbf{x}) \cdot u \cdot \zeta(\nabla g(\mathbf{x}))^{-1})$. Analogously, target points can be mapped back to the source domain using $(\mathbf{y}, v) \mapsto (\nabla f(\mathbf{y}), \zeta(\mathbf{y}) \cdot v \cdot \eta(\nabla f(\mathbf{y}))^{-1})$.

**Recovering semi-couplings.** Let us define $\tilde{\Gamma}_1 \overset{\text{def.}}{=} \text{diag}(\mathbf{e}^{-1})^\top \Gamma_1$ and $\tilde{\Gamma}_2 \overset{\text{def.}}{=} \text{diag}(\mathbf{z}^{-1})^\top \Gamma_2$, where $\Gamma_1, \Gamma_2$ are the solutions of the UBOT problems computed in Algorithm 1 (lines 7 and 9, respectively). It is easy to see that $(\tilde{\Gamma}_1, \tilde{\Gamma}_2^\top)$ is a valid pair of semi-couplings between $\mu$ and $\nu$ (cf. Eq. 8).

## 4   EVALUATION

We illustrate the effectiveness of NUBOT on different tasks, including a synthetic setup as well as an important but challenging task to predict single-cell perturbation responses to a diverse set of cancer drugs with different modes of actions.

Figure 2: **Unbalanced sample mapping**. In all three scenarios (a, b, c), the source (gray) and target (blue) datasets share structure but have different shifts and per-cluster sampling proportions. The true growth factors of the clusters depicted in the left pane bottom for each setup (True Weights). Tasked with mapping from source to target, NUBOT and UBOT GAN predict the locations (middle pane, red) and weights (right pane) of the transported samples. The number next to the weights denotes the mean weights per cluster. While both methods map the samples to the correct location, NUBOT more accurately predicts the weights needed to match the target distribution, creating mass (dark blue) or destroying it (red) as needed.

**Baselines.** To put NUBOT's performance into perspective, we compare it to several baselines: First, we consider a balanced neural optimal transport method CELLOT (Bunne et al., 2021), based on the neural dual formulation of Makkuva et al. (2020). Further, we benchmark NUBOT against the current state-of-the-art UBOT GAN, an unbalanced OT formulation proposed by Yang & Uhler (2019), which simultaneously learns a transport map and a scaling factor for each source point in order to account for variation of mass. Additionally, we consider two naive baselines: IDENTITY, simulating the identity matching and modeling cell behavior in absence of a perturbation, and OBSERVED, a random permutation of the observed target samples and thus a *lower bound* when comparing predictions to observed cells on the distributional level. Further, we consider DISCRETE OT, the entropy-regularized Wasserstein mapping returned by the Sinkhorn algorithm on finite source and target samples. As this method does not parameterize the transport map, we need to include the test cells while computing the optimal coupling and can therefore not be considered out-of-sample. GAUSSIAN APPROX computes a Gaussian approximation of the source and target samples separately, and uses the closed-form solution of the entropy-regularized optimal transport problem on unbalanced Gaussians (Janati et al., 2020b) for mapping. More details can be found in the Appendix §D.2.

## 4.1    SYNTHETIC DATA

Populations are often heterogeneous and consist of different subpopulations. To simulate such heterogeneous intervention responses which exhibit changes in their particle counts, we generate a dataset containing a two-dimensional mixture of Gaussians with three clusters in the source distribution $\mu$. The target distribution $\nu$ consists of the same three clusters, but with different cluster proportions. Further, each particle has undergone a constant shift in space upon intervention. We consider three scenarios with increasing imbalance between the three clusters (see Fig. 2a-c). We evaluate NUBOT on the task of predicting the distributional shift from source to target, while at the same time correctly rescaling the clusters such that no mass is transported across non-corresponding clusters.

**Results.** The results (setup, predicted Monge maps and weights) are displayed in Fig. 2. Both NUBOT and UBOT GAN correctly map the points to the corresponding target clusters without transporting mass across clusters. NUBOT also accurately models the change in cluster sizes by predicting the correct weights for each point. In contrast, UBOT GAN only captures the general trend of cluster growth and shrinkage, but does not learn the exact weights required to re-weight the cluster proportions appropriately. The exact setup and calculation of weights can be found in §B (see Table 1), as well as an evaluation of the robustness of NUBOT w.r.t. several hyperparamaters (see Fig. 7) and a comparison to UBOT (see Fig. 8, 9).

## 4.2    SINGLE-CELL PERTURBATION RESPONSES

Through the measurement of genomic, transcriptomic, proteomic or phenotypic profiles of cells, and the identification of different cell types and cellular states based on such measurements, biologists

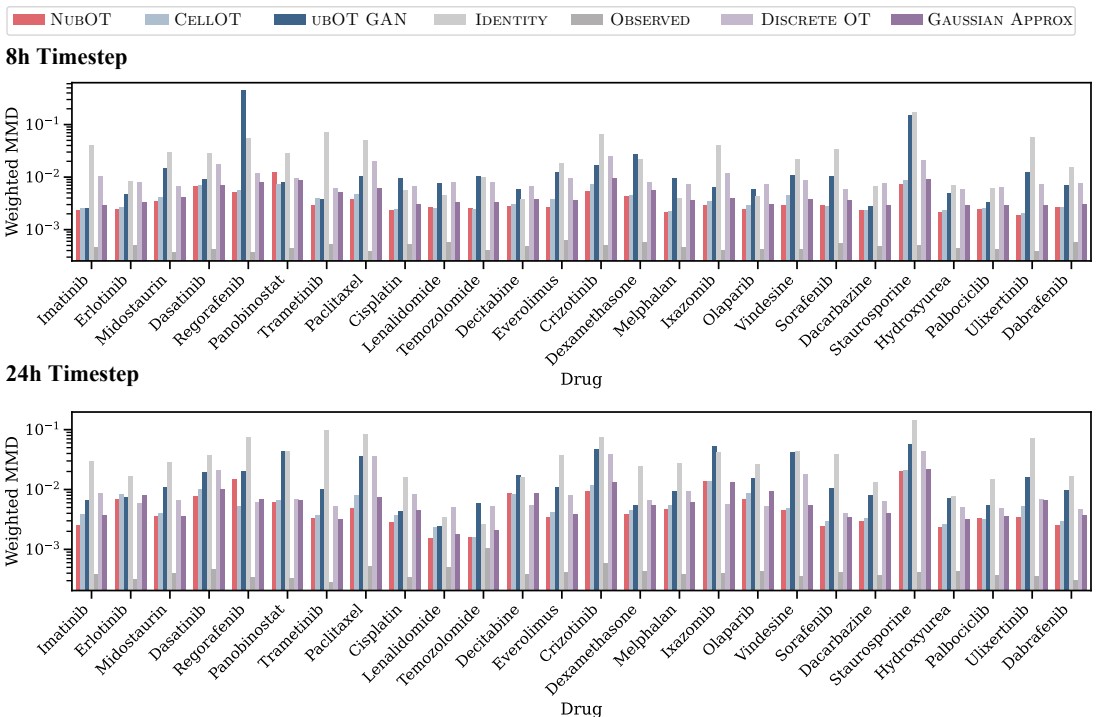

Figure 3: Distributional fit of the predicted perturbed cell states to the observed perturbed cell states for each drug and timestep, measured by a weighted version of kernel MMD on a set of held-out control cells. For NUBOT and UBOT GAN, MMD is weighted by the predicted weights, while for the other baselines it is computed with uniform weights. OBSERVED corresponds to a random permutation of the observed control cells, i.e., its distribution is approximately the same as the observed cells.

have revealed perturbation responses and response mechanism which would have remained obscured in bulk analysis approaches (Green & Pelkmans, 2016; Liberali et al., 2014; Kramer et al., 2022). However, single-cell measurements typically require the destruction the cells in the course of recording.

Thus, each measurement provides us only with a *snapshot* of cell populations, i.e., samples of probability distribution that is evolving in the course of the perturbation, from control $\mu$ (source) to perturbed cell states $\nu$ (target). Using NUBOT and the considered baselines, we learn a map $T$ that reconstructs how individual cells respond to a treatment. The effect of a single perturbation frequently varies depending on the cell type or cell state, and may include the induction of cell death or proliferation. In the following, we will evaluate if NUBOT is able to capture and predict heterogeneous proliferation and cell death rates of two co-cultures melanoma cell liens through $\eta$ and $\zeta$ in response to 25 drug treatments.

The single-cell measurements used for this task were generated using the imaging technology 4i (Gut et al., 2018) over the course of 24 hours, resulting in three different unaligned snapshots ($t = 0h$, $t = 8h$ and $t = 24h$) for each of the drug treatments. The control cells, i.e., the source distribution $\mu$, consists of cells taken from a mixture of melanoma cell lines at $t = 0h$ that are exposed to a dimethyl sulfoxide (DMSO) as a vehicle control. Futher, We consider two different target populations $\nu$ capturing the perturbed populations after $t = 8h$ and $t = 24h$ of treatment, respectively. As both cancer cell lines exhibit different sensitivities to the drugs (Raaijmakers et al., 2015), their proportion (Fig. 15) as well as the total cell counts (Fig. 17) vary over the time points. Both cell lines are characterized by the expression of mutually exclusive protein markers, i.e., one cell line strongly expresses a set of proteins detected by an antibody called MelA (MelA$^+$ cell type), while the other is characterized by high levels of the protein Sox9 (Sox9$^+$ cell type). An evaluation of this cell line annotation can be found in Fig. 14 (8h) and Fig. 16 (24h). As no ground truth matching is available, we use insights from the number of cells after 8 and 24 hours of treatment (Fig. 15, 17), as well as the cell type annotation for each cell to further evaluate NUBOT's performance. A detailed description of the dataset can be found in § C.2.

**Results.** We split the dataset into a train and test set and train NUBOT as well as the baselines on unaligned unperturbed (control) and perturbed cell populations for each drug. During evaluation, we then predict out-of-sample the perturbed cell state from held-out control cells. Details on the network architecture and hyperparameters can be found in § D.3. NUBOT and UBOT GAN additionally predict the weight associated with the perturbed predicted cells, giving insights into which cells have proliferated or died in response to the drug treatments. First, we compare how well each method fits the observed perturbed cells on the level of the entire distribution. For this, we measure the weighted version of kernel maximum mean discrepancy (MMD) between predictions and observations. More details on the evaluation metrics can be found in § D.1. The results are displayed in Fig. 3. We additionally report the weighted Wasserstein distance in Fig. 10 in Appendix §B. NUBOT outperforms all baselines in almost all drug perturbations, showing its effectiveness in predicting OT maps and local variation in mass.

In the absence of a ground truth and in particular, given our inability to measure (i.e., observe) cells which have died upon treatment, we are required to base further analysis of NUBOT's predictions on changes in cell count for each subpopulation (MelA$^+$, Sox9$^+$). Fig. 15 clearly shows that drug treatments lead to substantially different cell numbers for each of the subpopulations compared to control. For example, Ulixertinib leads to the proliferation of both subpopulations after 8h, but to pronounced cell death in Sox9$^+$ and strong proliferation in MelA$^+$ cells after 24h. We thus expect, that weights predicted by NUBOT for all drugs correlate with the change in cell counts for each cell type (here measured as population fractions). This is indeed the case, Fig. 4 shows a high correlation between observed cell counts of the two cell types and the sum of the predicted weights of the respective cell types after 8h of treatment for all drugs. After 24 hours, treatment-induced cell death (in at least one cell type) by some drugs can be so severe at that the number of observed perturbed cells becomes too low for accurate predictions and the evaluation of the task (Fig. 17). Further, we find that drugs influence the abundance of the cell lines markers MelA and Sox9, complicating cell type classification (see Fig. 14, 16). We ignore drugs falling into these categories and find that whilst the correlation between predicted weights and observed cell counts is reduced after 24h (see Fig. 4a), NUBOT still captures the overall trend.

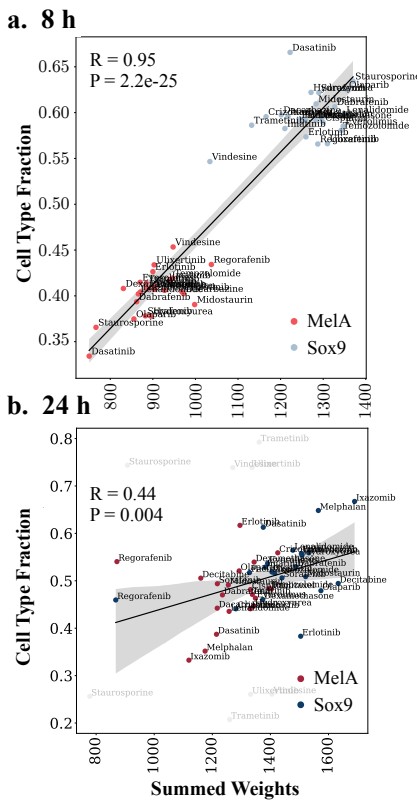

Figure 4: Given the ground truth on the known subpopulation (MelA (red) and Sox9 (blue)) sizes for each drug, we analyze their level of correlation to our predicted weights after **a.** 8h and **b.** 24h. With increasing difficulty of the task and certain drugs completely removing both or one of the subpopulations, the level of correlation reduces from 8 to 24h. $R$ denotes Pearson correlation coefficient and $P$ the p-value.

The data further provides insights into biological processes such as apoptosis, a form of programmed cell death induced by enzymes called Caspases (ClCasp3). While dead cells become invisible in the cell state space (they cannot be measured), *dying* cells are still present in the observed perturbed sample and can be recognized by high levels of ClCasp3 (the apopotosis markers). Conversely, the protein Ki67 marks proliferating cells. Analyzing the correlation of ClCasp3 and Ki67 intensity with the predicted weights provides an additional assessment of the biological meaningfulness of our results. For example, upon Ulixertinib treatment, the absolute cell counts show an increase in Sox9$^+$ cells, and a decline of MelA$^+$ cells at 24h (Fig. 15). Fig. 5 shows UMAP projections of the control cells at both time points, colored by the observed and predicted protein marker values and the predicted weights. At 8h, NUBOT predicts only little change in mass, but a few proliferative cells with high weights in areas which are marked by high values of the proliferation marker Ki67. At 24h, our model predicts cell death in the Sox9$^+$ (MelA$^-$) cell type, and proliferation in the MelA$^+$ cell type, which matches the observed changes in cell counts per cell type, seen in Fig. 15 in § B. We identify similar results for Trametinib (Fig. 11), Ixazomib (Fig. 12), and Vindesine (Fig. 13) which can be found in § B. These experiments thus demonstrate that NUBOT accurately predicts heterogeneous drug responses at the single-cell level, capturing both, cell proliferation and death.

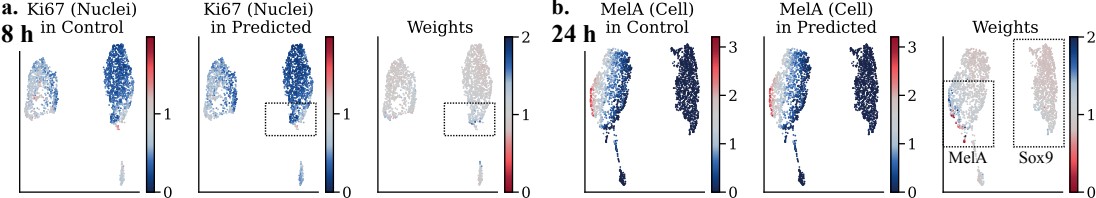

Figure 5: UMAP projections of the control cells for Ulixertinib at **a.** 8h and **b.** 24h. Cells are colored by the observed and predicted protein marker values (Ki67, MelA), and predicted weights. NUBOT thereby correctly predicts weights $\geq 1$ for proliferating cells in the MelA$^+$ population (**a.** and **a.**, right panel), and increased levels of cell death in the Sox9$^+$ population after 24h via weights $\leq 1$ (**b.**, right panel), confirmed by the experimental observations (see Fig. 15).

## 5 CONCLUSION

This work presents a novel formulation of the unbalanced optimal transport problem that bridges two previously disjoint perspectives on the topic: a theoretical one based on semi-couplings and a practical one based on recent neural estimation of OT maps. The resulting algorithm, NUBOT, is scalable, efficient, and robust. Yet, it is effective at modeling processes that involve population growth or death, as demonstrated through various experimental results on both synthetic and real data. On the challenging single-cell perturbation task, NUBOT is able to successfully predict perturbed cell states, while explicitly modeling death and proliferation. Explicitly modeling proliferation and death at the single-cell level as part of the drug response, allows to link cellular properties observed prior to drug treatment to therapy outcomes. Thus, the application of NUBOT in the fields of drug discovery and personalized medicine could be of great implications, as it allows to identify cellular properties predictive of drug efficacy.

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

# APPENDIX

## A    RELATED WORK

In the following, we provide further information and review related literature on concepts discussed throughout this work.

### A.1    UNBALANCED OPTIMAL TRANSPORT

Unbalanced optimal transport is a generalization of the classical OT formulation (1), and as such allows mass to be created and destroyed throughout the transport. This relaxation has found recent use cases in various domains ranging from biology (Schiebinger et al., 2019; Yang & Uhler, 2019), imaging (Lee et al., 2019), shape registration (Bonneel & Coeurjolly, 2019), domain adaption (Fatras et al., 2021), positive-unlabeled learning (Chapel et al., 2020), to general machine learning (Janati et al., 2020a; Frogner et al., 2015). Problem (6) provides a general framework of the unbalanced optimal transport problem, which can recover related notions introduced in the literature: Choosing for $\mathcal{D}_f$ the Kullback-Leibler divergence, one recovers the so-called squared Hellinger distance, with $\mathcal{D}_f = \ell_2$ norm, we arrive at Benamou (2003), and an $\ell_1$ norm retrieves a concept often referred to robust OT (Mukherjee et al., 2020) with connections to a concept known as partial OT (Figalli, 2010). The latter comprises approaches which do not rely on a relaxation of the marginal constraints as in (6). In particular, some strategies of partial OT expand the original problem by adding *virtual* mass to the marginals (Pele & Werman, 2009; Caffarelli & McCann, 2010; Gramfort et al., 2015), or by extending the OT map by *dummy* rows and columns (Sarlin et al., 2020) onto which excess mass can be transported. The semi-coupling formulation (also known as Wasserstein-Fisher-Rao distance) has been independently proposed by Liero et al. (2016; 2018), and recently been connected to shape analysis of surfaces (Bauer et al., 2022). Recent work has furthermore developed alternative computational schemes (Chapel et al., 2021; Séjourné et al., 2022b) as well as provided a computational complexity analysis (Pham et al., 2020) of the generalized Sinkhorn algorithm solving entropic regularized unbalanced OT (Chizat et al., 2018a). Besides Yang & Uhler (2019), these approaches do not provide parameterizations of the unbalanced problem and allow for an out-of-sample generalization which we consider in this work. The notion of unbalanced optimal transport has been further generalized to the multi-marginal setting Beier et al. (2022) as well as for Gromov-Wasserstein (Séjourné et al., 2021). Janati et al. (2020b) further provide a closed-form solution of entropic optimal transport between unbalanced Gaussian measures. For a complete review, we refer the reader to Séjourné et al. (2022a) as well as (Peyré & Cuturi, 2019, Chapter 10.2).

It becomes evident from the discussion above that unbalanced optimal transport is better thought of as a *family* of problems, rather than a single specific one. Unbalanced OT arises in any setting where one seeks a notion of distance or correspondence across datasets/populations *and* (i.) the populations are not normalized in the same way and/or (ii.) one does not want to rigidly enforce full correspondence between the datasets (e.g., to minimize the effect of outliers or because there is reason to believe some datapoints do not occur in both datasets). This work proposes a method to parameterize an UBOT problem which does not necessarily match the typical Kantorovich entropy-regularized formulation. We seek a solution that learns which part of the distribution shrinks and which one exhibits overall growth, motivated by heterogeneous sobpopulation structures present in single-cell biology. To that end, NUBOT aims at modeling birth and death dynamics throughout the distribution shift recovered through an optimal transport mapping.

### A.2    CYCLE-CONSISTENT LEARNING

The principle of cycle-consistency has been widely used for learning bi-directional transformations between two domains of interest. Cycle-consistency thereby assumes that both the forward and backward mapping are roughly inverses of each other. In particular, given unaligned points $x \in \mathcal{X}$ and $y \in \mathcal{Y}$, as well as maps $g : \mathcal{X} \mapsto \mathcal{Y}$ and $f : \mathcal{Y} \mapsto \mathcal{X}$, cycle-consistency reconstruction losses enforce $\|x - f(g(x))\|$ as well as $\|y - g(f(y))\|$ using some notion of distance $\| \cdot \|$, assuming that there exists such a ground truth bijection $g = f^{-1}$ and $f = g^{-1}$. The advantage of validating *good* matches by cycling between *unpaired* samples becomes evident through the numerous use cases to which cycle-consistency has been applied: Originally introduced within the field of computer vision

(Kalal et al., 2010) and applied to image-to-image translation tasks (Zhu et al., 2017a), it has been quickly adapted to multi-modal problems (Zhu et al., 2017b), domain adaptation (Hoffman et al., 2018), and natural language processing (Shen et al., 2017). The original principle has been further generalized to settings requiring a many-to-one or surjective mapping between domains (Guo et al., 2021) via conditional variational autoencoders, dynamic notions of cycle-consistency (Zhang et al., 2021), or to time-varying applications (Dwibedi et al., 2019). These classical approaches enforce cycle-consistency by *explicitly* composing both maps and penalizing for any deviation from this bijection. In this work, we treat cycle-consistency differently. It is enforced implicitly by coupling the two distributions of interest through a sequence of reversible transformations: re-weighting, transforming, and re-weighting (Eq. (9) and Fig. 1).

Similarly to our work, Zhang et al. (2022) and Hur et al. (2021) establish a notion of cycle-consistency (reversibility) for a pair of pushforward operators to align two unpaired measures. Both methods rely on the Gromov-Monge distance (Mémoli & Needham, 2022), a divergence to compare probability distributions defined on different ambient spaces $\mathcal{X}$ and $\mathcal{Y}$—a setting not considered in this work. They proceed by defining a reversible metric through replacing the single Monge map by a pair of two Monge maps, i.e., $f : \mathcal{X} \mapsto \mathcal{Y}$ and $g : \mathcal{Y} \mapsto \mathcal{X}$, minimizing the objective

$$\mathrm{GM}(\mu, \nu) := \inf_{\substack{f:\mathcal{X}\mapsto\mathcal{Y}, f_\sharp\mu=\nu \\ g:\mathcal{Y}\mapsto\mathcal{X}, g_\sharp\nu=\mu}} \Delta_{\mathcal{X}}^p(f; \mu) + \Delta_{\mathcal{Y}}^p(g; \nu) + \Delta_{\mathcal{X},\mathcal{Y}}^p(f, g; \mu, \nu), \tag{14}$$

$$\Delta_{\mathcal{X}}^p(f; \mu) = \left(\mathbb{E}\left[|c_{\mathcal{X}}(x, x') - c_{\mathcal{Y}}(f(x), f(x'))|^p\right]\right)^{\frac{1}{p}}$$

$$\Delta_{\mathcal{Y}}^p(g; \nu) = \left(\mathbb{E}\left[|c_{\mathcal{X}}(y, y') - c_{\mathcal{Y}}(g(y), g(y'))|^p\right]\right)^{\frac{1}{p}}$$

$$\Delta_{\mathcal{X},\mathcal{Y}}^p(f, g; \mu, \nu) = \left(\mathbb{E}\left[|c_{\mathcal{X}}(x, g(y)) - c_{\mathcal{Y}}(f(x), y)|^p\right]\right)^{\frac{1}{p}}.$$

Problem (14) shows similarities to the classical cycle-consistency objective of Zhu et al. (2017a), where cycle-consistency is indirectly enforced through $\Delta_{\mathcal{X},\mathcal{Y}}^p$. Zhang et al. (2022) parameterize both Monge maps through neural networks in a similar fashion as done in (Yang & Uhler, 2019; Fan et al., 2021a). Our approach differs from Zhang et al. (2022); Hur et al. (2021) as we model the problem through a single Monge map with duals $f, g$, allowing us to map back-and-forth between measures $\mu$ and $\nu$, and using a different parametrization approach (ICNNs). More importantly, the approaches presented by Zhang et al. (2022); Hur et al. (2021) do not generalize to the unbalanced case. While Zhang et al. (2022) proposed an unbalanced version of (14) by relaxing the marginals as done in Chizat et al. (2018a), they require the unbalanced sample sizes to be known (i.e., $n$ and $m$ need to be fixed). In our application of interest, particle counts of the target population are, however, not known *a priori*.

### A.3    CONVEX NEURAL ARCHITECTURES

Input convex neural networks (Amos et al., 2017) are a class of neural networks that approximate the family of convex functions $\psi$ with parameters $\theta$, i.e., whose outputs $\psi_\theta(x)$ are convex w.r.t. an input $x$. This property is realized by placing certain constraints on the networks parameters $\theta$. More specifically, an ICNN is an $L$-layer feed-forward neural network, where each layer $l = \{0, ..., L-1\}$ is given by

$$z_{l+1} = \sigma_l(W_l^x x + W_l^z z_l + b_l) \text{ and } \psi_\theta(x) = z_L, \tag{15}$$

where $\sigma_k$ are convex non-decreasing activation functions, and $\theta = \{W_l^x, W_l^z, b_l\}_{l=0}^{L-1}$ is the set of parameters, with all entries in $W_l^z$ being non-negative and the convention that $z_0$ and $W_0^z$ are 0. As mentioned above and through the connection established in § 2, convex neural networks have been utilized to approximate Monge map $T$ (3) via the convex Brenier potential $\psi$ connected to the primal and dual optimal transport problem. In particular, it has been used to model convex dual functions (Makkuva et al., 2020) as well as normalizing flows derived from convex potentials (Huang et al., 2021). The expressivity and universal approximation properties of ICNNs have been further studied by Chen et al. (2019), who show that any convex function over a compact convex domain can be approximated in sup norm by an ICNN. To improve convergence and robustness of ICNNs —known to be notoriously difficult to train (Richter-Powell et al., 2021)— different initialization schemes have been proposed: Bunne et al. (2022b) derive two initialization schemes ensuring that *upon initialization* $\nabla\psi$ mimics an affine Monge map $T$ mapping either the source measure onto itself

Table 1: Setup of the synthetic mixture of Gaussians dataset, showing the proportions of the three clusters in source and target distribution in three different settings (**a.**, **b.**, **c.**) as well as the required scaling factor per cluster needed to match the target without transporting points to non-corresponding clusters. The last two columns show the mean weights obtained by NUBOT and UBOT GAN.

| Setting | Cluster | True Scaling Factor | Mean Weights | | Difference to Ground Truth | |
|---------|---------|---------------------|--------------|---------|----------------------------|---------|
| | | | **NubOT** | **ubOT GAN** | **NubOT** | **ubOT GAN** |
| **a.** | 1 | 0.3 | 0.26 | 0.8 | **0.04** | 0.5 |
| | 2 | 1.35 | 1.36 | 0.99 | **0.01** | 0.36 |
| | 3 | 1.35 | 1.32 | 1.02 | **0.03** | 0.33 |
| **b.** | 1 | 0.3 | 0.29 | 0.81 | **0.01** | 0.51 |
| | 2 | 0.6 | 0.64 | 0.88 | **0.04** | 0.28 |
| | 3 | 2.1 | 2.08 | 1.18 | **0.02** | 0.92 |
| **c.** | 1 | 4.5 | 4.6 | 1.44 | **0.1** | 3.06 |
| | 2 | 1.0 | 0.98 | 0.94 | **0.02** | 0.06 |
| | 3 | 0.22 | 0.23 | 0.79 | **0.01** | 0.57 |

(identity initialization) or providing a map between Gaussian approximations of measures $\mu$ and $\nu$ (Gaussian initialization). Further, Korotin et al. (2020) proposed to use quadratic layers as well as a pre-training pipeline to initialize ICNN parameters to encode an identity map.

# B    ADDITIONAL EXPERIMENTAL RESULTS

## B.1    SYNTHETIC DATA

In our synthetic two-dimensional dataset, the source and target distribution are mixtures of Gaussians with varying proportions. Both source and target consist of three corresponding clusters, and by changing the proportions of each cluster, we illustrate a scenario in which subpopulations grow and shrink at different rates. The clusters are sampled according to the following probabilities: **a.** source $\{0.3, 0.3, 0.3\}$ to target $\{0.1, 0.45, 0.45\}$, **b.** source $\{0.3, 0.3, 0.3\}$ to target $\{0.1, 0.2, 0.7\}$, **c.** source $\{0.1, 0.45, 0.3\}$ to target $\{0.45, 0.45, 0.1\}$. Table 1 shows the shares of the three clusters in the source and target distributions. In order to match the target distribution without transporting mass across non-corresponding clusters, the clusters have to be re-scaled with the factors presented in column 'True Scaling Factor'. The last two columns show the mean weights per cluster obtained by NUBOT and UBOT GAN, respectively. UBOT GAN captures only the general trend in growth and shrinkage, the exact weights do not scale the cluster proportions appropriately. In contrast, the weights obtained by NUBOT match the required scaling factors very closely. Fig. 6, shows the weighted MMD between the source distribution and the target distribution, confirming superior performance of NUBOT.

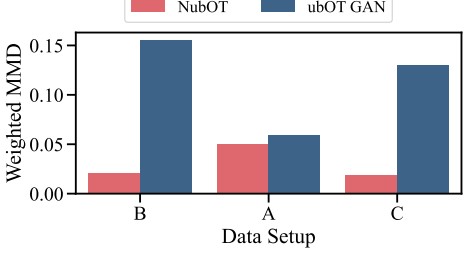

Figure 6: Distributional fit of the predicted samples to the target samples on synthetic data, measured by a weighted version of kernel MMD.

### B.1.1 ROBUSTNESS OF NUBOT W.R.T. HYPERPARAMETERS

In the following, we evaluate NUBOT's sensitivity to hyperparameter choices. For this we screen various parameter ranges of the batch size (`bs`), as well as hyperparameters of the generalized Sinhorn algorithm (Chizat et al., 2018a) to solve the UBOT (see Algorithm 1 l. 5 and 7), i.e., the entropy regularization parameters (`reg`), and relaxation penalties (`reg`$_m$). All considered hyperparameters can be found in the legend of Fig. 7. The base model thereby denotes the final hyperparameters chosen for all experiments conducted in this work. We conduct this analysis on the synthetic data setup introduced in § 4.1 and Figure 2a and b, i.e., a mixture of three Gaussians exhibiting different changes in particle count with known ground truth. We compare the obtained weights of each hyperparameter configuration for each setups (**a.** and **b.**). The results displayed in Fig. 7 (left panels) demonstrate that the correct weights are robustly learned for all (in setting **a.**) and most (in setting **b.**) hyperparameter choices. In setup **b.**, the weights deviate in particular for high values of `reg`$_m(\geq 0.5)$. Fig. 7 (right panels) further show the correlation between the obtained weights of the base model (i.e., the parameters chosen for all experiments) with all other hyperparameter configurations.

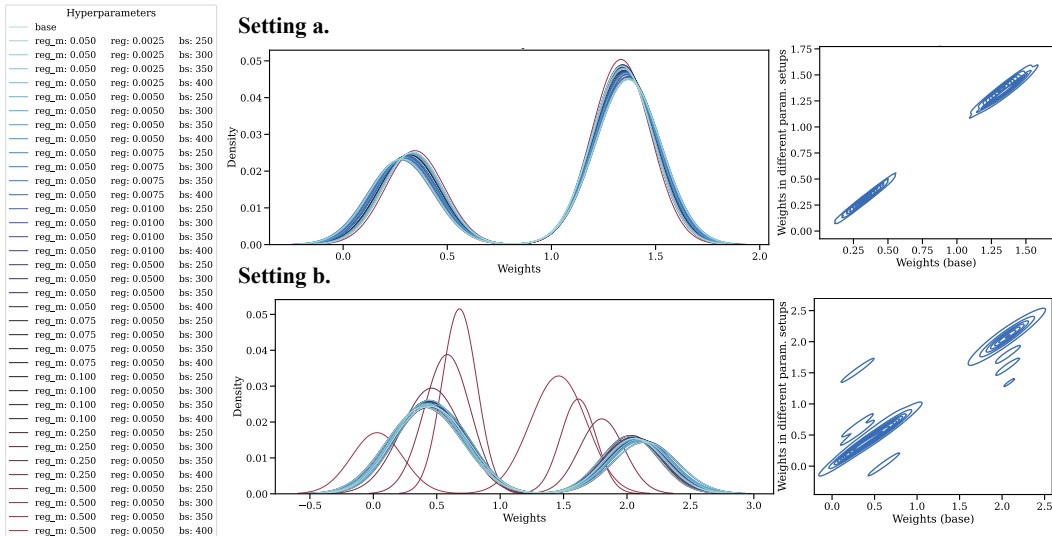

Figure 7: **Hyperparameter screen of the NUBOT model.** We screen various different hyperparameter configurations different to the base model (the one used for the experiments above) and evaluated them for different synthetic data settings (**a.** and **b.**). (left panels) Distribution of learned weights obtained for each hyperparameter setting and (right panels) correlation of the weights for all hyperparameter configurations compared to the base setup.

### B.1.2 COMPARISON TO UBOT

NUBOT provides a parameterization of an unbalanced optimal transport problem. At each training step, NUBOT executes the generalized Sinkhorn algorithm (Chizat et al., 2018a; Cuturi, 2013; Benamou et al., 2015) as a subroutine to obtain estimates of the weights **e** and **z**. To compare the solution of NUBOT to the mapping computed by the UBOT problem in (7), we further analyze the obtained solutions and hyperparameter sensitivities of UBOT itself, i.e., not integrated within NUBOT. In particular, we compare different relaxation penalties $\tau = \tau_1 = \tau_2$ (see (7)), and different choices of the entropy regularization $\epsilon$.

We consider the same synthetic data settings (**a.**, **b.**) of Gaussian mixtures, as introduced in § 4 and shown in Fig. 2, where the three clusters grow/shrink at different rates which are known. We show the couplings and weights obtained from UBOT in Fig. 8 (setting **a.**) and Fig. 9 (setting **b.**), for different regularization ($\epsilon$) and relaxation penalty ($\tau$) parameters. The weights per source point are thereby computed by summing over the columns in the coupling matrix. To get a weight value relative to all other weights, we additionally normalize these weights by the total sum of the coupling matrix. In most cases UBOT couples points only between corresponding clusters. As $\tau$ increases

for $\epsilon = 0.05$, mass variation is penalized stronger, and therefore, mass has to be coupled between non-corresponding clusters as well in order to fit the marginals.

The non-normalized weights exhibit a significantly higher variation between different values of $\tau$, especially for $\epsilon = 0.05$. Comparing the average weights per cluster to the true scaling, which are shown in Table 1 and Fig. 2, we observe that both the non-normalized and normalized weights are matching these values closely for $\tau \in \{1, 10\}$ and $\epsilon = 0.005$. For $\epsilon = 0.05$ and small values of $\tau$, the coupling is still only mapping points between corresponding clusters. The weights, however, are not matching the true underlying scalings.

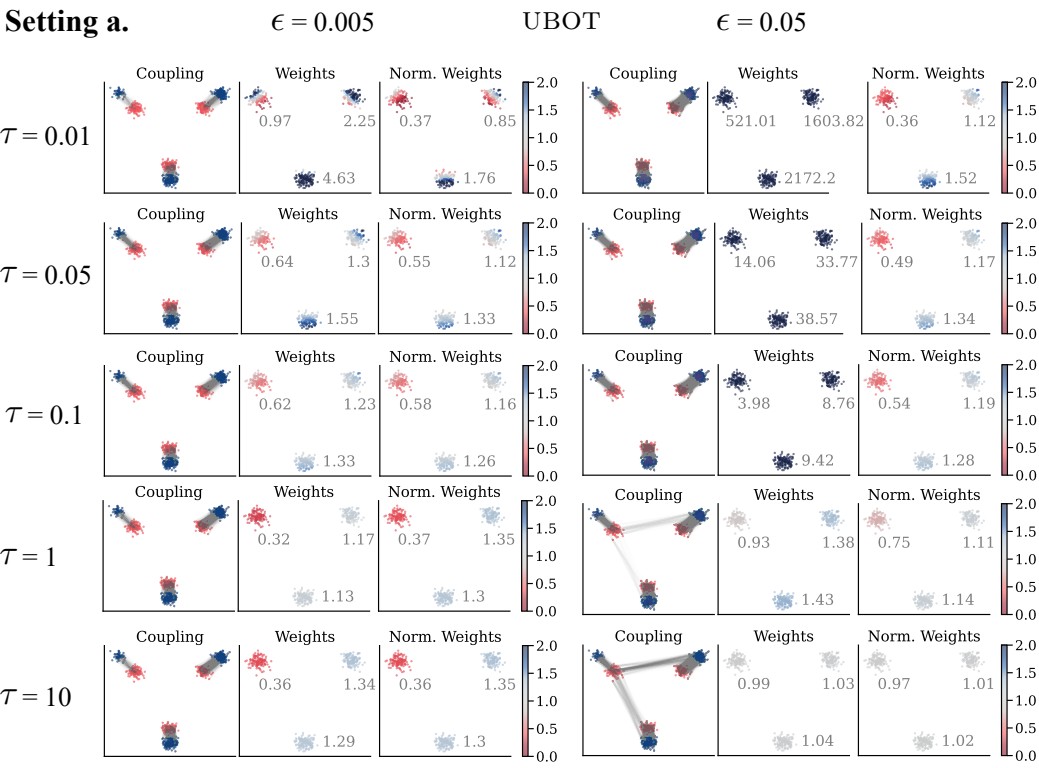

Figure 8: Discrete solution to synthetic experiments in setting **a.** using unbalanced Sinkhorn (UBOT), with different entropy regularization parameters $\epsilon$ and different penalization parameters $\tau = \tau_1 = \tau_2$ (7). For each parameter setup, we show the coupling from source (red) to target (blue), where the strength of each line is proportional to its value in the coupling matrix (left pane), the weights (middle pane) and the normalized weights (right pane). The weight per point is computed as the sum over the columns of the coupling matrix. We additionally show the average weight per cluster.

## B.2    SINGLE-CELL PERTURBATION RESPONSES

In addition to the weighted MMD metric shown in Fig. 3, we evaluated our method on another distributional metric, the weighted Wasserstein distance between the predicted perturbed and observed perturbed cells. We compute it with the Sinkhorn algorithm, whereby for NUBOT and UBOT GAN, we pass the normalized predicted weights as source weights. Results are shown in Fig. 10.

As we lack ground truth for the correspondence of control and perturbed cells, we assess the biological meaningfulness of our predictions by comparing the weights to ClCasp3 and Ki67 intensity, the apoptosis and proliferation markers, respectively. Figures 11, 12 and 13 show UMAP projections computed on control cells for the drugs Trametinib, Ixazomib and Vindesine. In Figure 12 c., d., and Figure 13 c., d., regions of low predicted weights accurately correspond to regions of increased ClCasp3 intensity. Additionally, we compare predicted weights between the two cell types, and contrast them with observed cell counts.

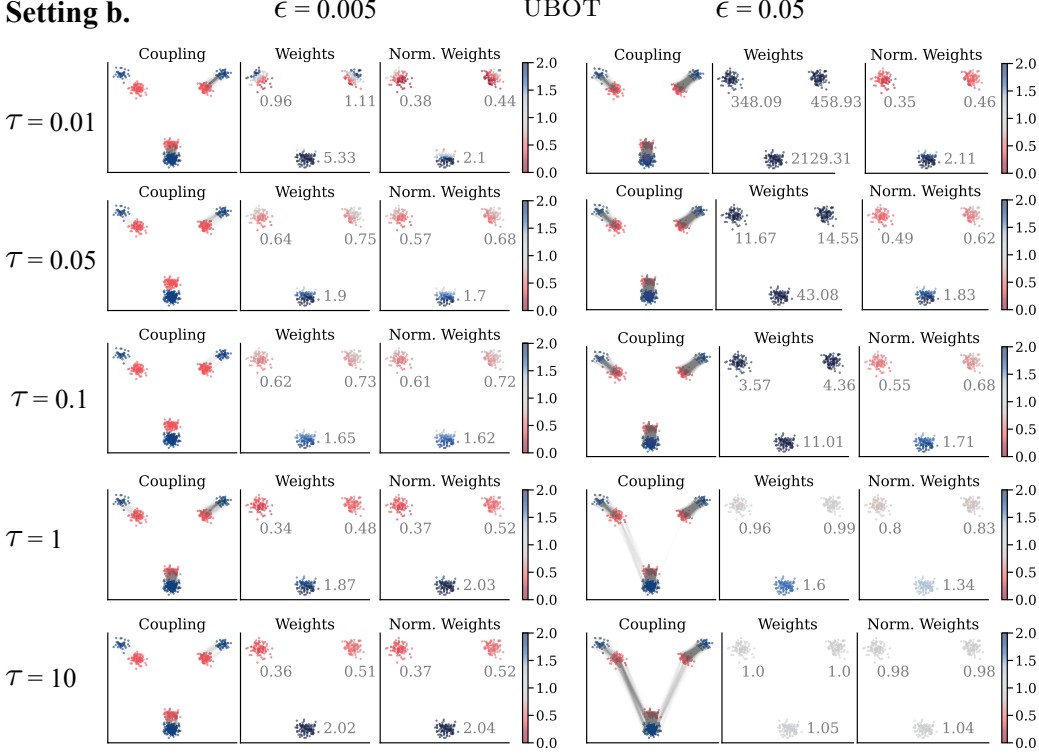

Figure 9: Discrete solution to synthetic experiments in setting **b.** using unbalanced Sinkhorn (UBOT), with different entropy regularization parameters $\epsilon$ and different penalization parameters $\tau = \tau_1 = \tau_2$ (7). For each parameter setup, we show the coupling from source (red) to target (blue), where the strength of each line is proportional to its value in the coupling matrix (left pane), the weights (middle pane) and the normalized weights (right pane). The weight per point is computed as the sum over the columns of the coupling matrix. We additionally show the average weight per cluster.

## C   DATASETS

We evaluate NUBOT on several tasks including synthetic data as well as perturbation responses of single cells. In both settings, we are provided with unpaired measures $\mu$ and $\nu$ and aim to recover map $T$ which describes how source $\mu$ transforms into target $\nu$. While in the synthetic data setting we are provided with a ground truth matching, this is not the case for the single-cell data as measuring a cell requires destroying it. In the following, we describe generation and characteristics of both datasets, as well as introduce additional biological insights allowing us to shed light on the learned matching $T$.

### C.1   SYNTHETIC DATA

To evaluate NUBOT in a simple and low-dimensional setup with known ground-truth, we generate synthetic example: We model a source population with clear subpopulation structure through a mixture of Gaussians. Next, we generate a second (target) population aligned to the source population. We then simulate an intervention to which the subpopulations respond differently, including different levels of growth and death. Specifically, we generate batches of 400 samples with three clusters with different proportions before and after the intervention. Table 1 shows the proportions of the three clusters in the source and target distribution, as well as the required weight-factor and the obtained results from NUBOT and UBOT GAN.

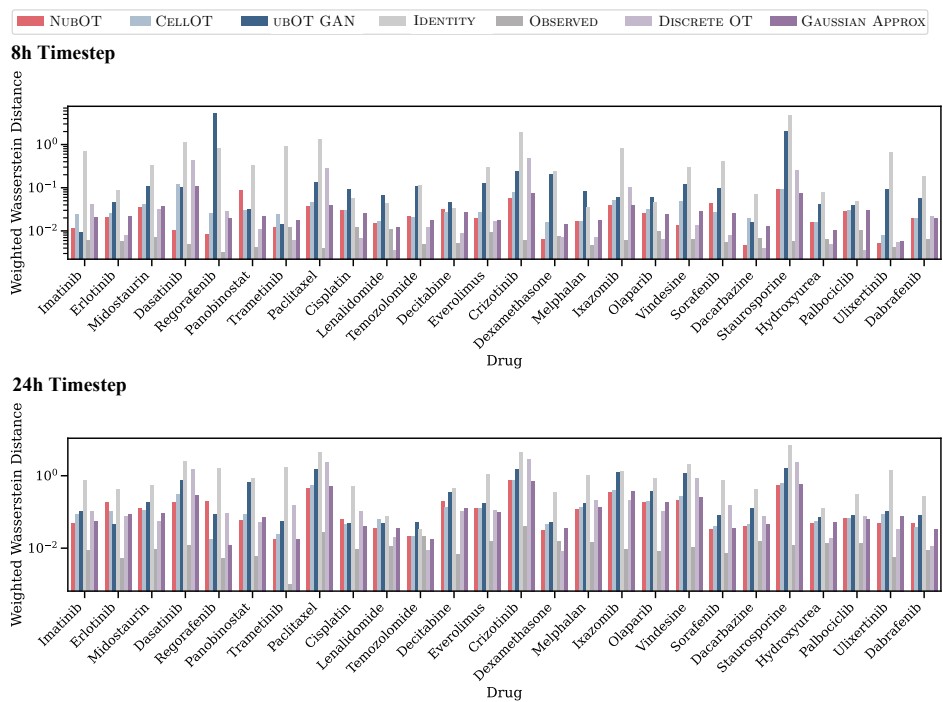

Figure 10: Distributional fit of the predicted perturbed cell states to the observed perturbed cell states for each drug and timestep, measured by the Wasserstein distance (2) on a set of held-out control cells. Here, histogram $\mathbf{u}$ in (2) of the predictions contains the learned weights (for those methods returning an unbalanced solution, i.e., NUBOT and UBOT GAN). For the other baselines $\mathbf{u}$ contains uniform weights.

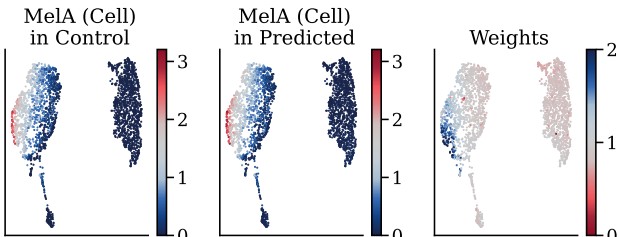

Figure 11: UMAP projections computed on control cells for Trametinib at $t = 24h$. High predicted weights in the MelA$^+$ cell type suggest proliferation, while the Sox9$^+$ population shows higher levels of cell death. This prediction is confirmed by the relative cell counts, where MelA$^+$ cell counts increase and Sox9$^+$ counts decrease, demonstrating opposite response behaviors to Trametinib for each subpopulation, i.e., MelA$^+$ cells show proliferation and Sox9$^+$ cells death.

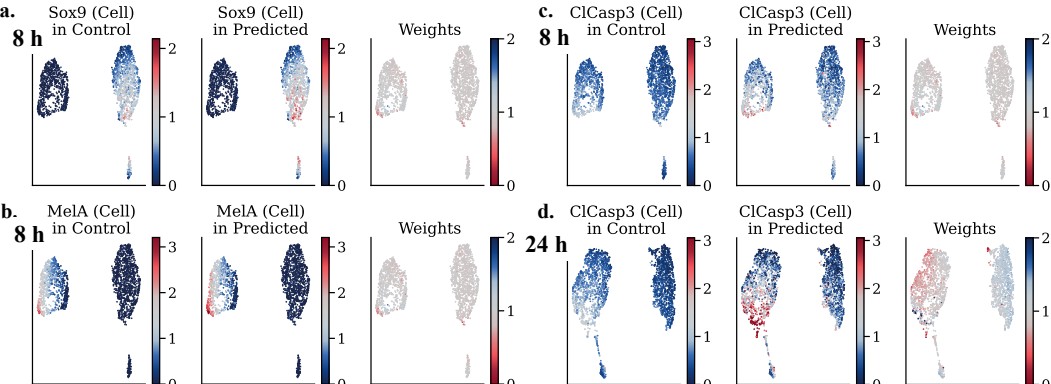

Figure 12: UMAP projections computed on control cells for Ixazomib for $t = 8h$, and $t = 24h$, colored by protein marker intensities **a.** MelA and **b.** Sox9, markers for the two subpopulations, as well as ClCasp3, a marker for cell death, at **c.** 8h and **d.** 24h. The UMAPs confirm the measured relative cell counts of each subpopulation. After 8h **a.-c.**, neither MelA$^+$ nor Sox9$^+$ cells are affected by the treatment, i.e., we mainly predict weights around 1. **d.** After 24h, we observe low weights in regions of high predicted apoptosis marker intensities (ClCasp3), especially at $t = 24h$, where the observed cell counts suggest death predominantly in the MelA$^+$ cell cluster.

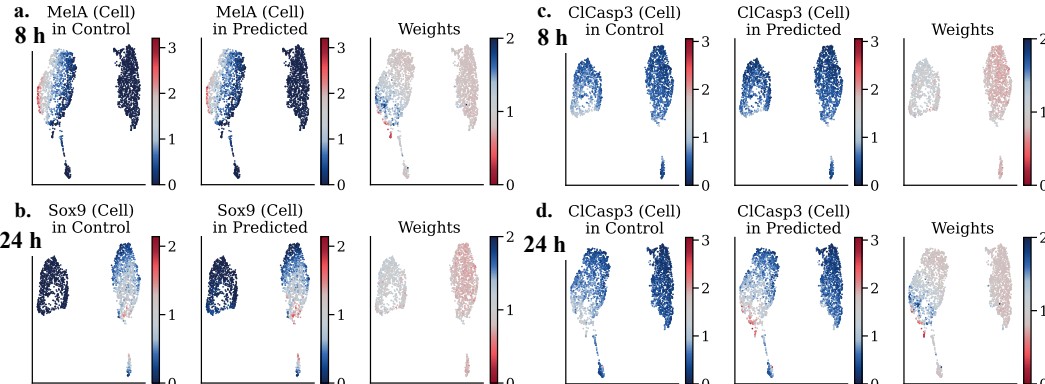

Figure 13: UMAP projections computed on control cells for Vindesine for $t = 8h$, and $t = 24h$, colored by protein marker intensities **a.** MelA and **b.** Sox9, markers for the two subpopulations, as well as ClCasp3, a marker for cell death, at **c.** 8h and **d.** 24h. The predicted weights (left) at **c.** 8h and **d.** 24h match the observed effects on each subpopulation, as initially only Sox9$^+$ cells are affected by treatment with Vindesine, and only after 24h MelA$^+$ cells show increased cell death.

Table 2: **Overview of all treatments and their inhibition type considered in this work.** Abbreviations PROTi (Proteasome inhibitor), DNASynthi (DNA synthesis inhibitor), panKi (pan kinase inhibitor), ImmuneMod. (Immune modulatory compound), MTDisruptor (Microtubule disruptor), ApopInducer (Apoptosis inducer).

| Drug Name | Inhibitor Type | Drug Name | Inhibitor Type |
|---|---|---|---|
| Ixazomib | PROTi | Olaparib | PARPi |
| Sorafenib | RAFi | Paclitaxel | MTDisruptor |
| Dabrafenib | BRAFi | Melphalan | Alkylator |
| Everolimus | mTORi | Regorafenib | panKi |
| Hydroxyurea | DNASynti | Vindesine | MTDisruptor |
| Midostaurin | panKi | Cisplatin | Alkalyting |
| Dexamethasone | ImmuneMod. | Ulixertinib | ERKi |
| Temozolomide | Alkylator | Staurosporine | ApopInducer |
| Decitabine | DNAMeti | Lenalidomide | ImmuneMod. |
| Dasatinib | SRCi-ABLi | Crizotinib | METi |
| Trametinib | MEKi | Imatinib | KITi-PDGFRi-ABLi |
| Erlotinib | EGFRi | Palbociclib | CDK4/6i |
| Dacarbazine | Alkylator | | |

## C.2    SINGLE-CELL DATA

**Biological experiment.**    The single-cell dataset used in this work was generated by the a multiplexed microscopy technology called Iterative Indirect Immunofluorescence Imaging (4i) (Gut et al., 2018), which is capable of measuring the abundance and localization of many proteins in cells. By iteratively adding, imaging and removing fluorescently tagged antibodies, a multitude of protein markers is captured for each cell. Additionally, cellular and morphological characteristics are extracted from microscopical images, such as the cell and nucleus area and circularity. This spatially resolved phenotypic dataset is rich in molecular information and provides insights into heterogeneous responses of thousands of cells to various drugs. Measuring different morphological and signaling features captures pre-existing cell-to-cell variability which might influence perturbation effect, resulting in various different responses. Some of these markers are of particular importance, as they provide insights into the level of a cell's growth or death as well as subpopulation identity. We utilized a mixture of two melanoma tumor cell lines (M130219 and M130429) at a ratio of 1:1. The cell lines can be differentiated by the mutually exclusive expression of marker proteins. The former is positive for Sox9, the latter for a set of four proteins which are all recognized by and antibody called MelA (Raaijmakers et al., 2015). Cells were seeded in a 384-well plate and incubated at 37C and 5% CO2 overnight. Next, the cells were exposed to multiple cancer drugs and Dimethyl sulfoxide (DMSO) as a vehicle control for 8h and 24h after which the cells were fixed and six cycles of 4i were performed TissueMAPS and the scikit-image library (Van der Walt et al., 2014) were used to process and analyze the acquired images, perform feature extraction and quality control steps using semi-supervised random forest classifiers.

**Data generation and processing.**    Our datasets contain high-dimensional single-cell data of control and drug-treated cells measured at two time points (8 and 24 hours). For both the 8h-dataset and the 24h-dataset, we normalized the extracted intensity and morphological features by dividing each feature by its 75th percentile, computed on the control cells. Additionally, values were transformed by a $log1p$ function ($x \leftarrow log(x + 1)$). In total, our datasets consist of 48 features, of which 26 are protein marker intensities and the remaining 22 are morphological features. For each treatment, we have measured between 2000 and 3000 cells. For training the models, we perform a 80/20 train/test split. We trained all models on control and treated cells for each time step and each drug separately. The considered drugs as well as their inhibition type can be found in Table 2.

**Cell type assignment.**    We assigned M130219 and M130429 cells to the Sox9 and MelA cell types, respectively, by first training a two component Gaussian mixture model on the features 'intensity-cell-MelA-mean' and 'intensity-nuclei-Sox9-mean' of the control cells. Next, we used the aforementioned features and the labels provided by the mixture model to train a nearest neighbor classifier, which

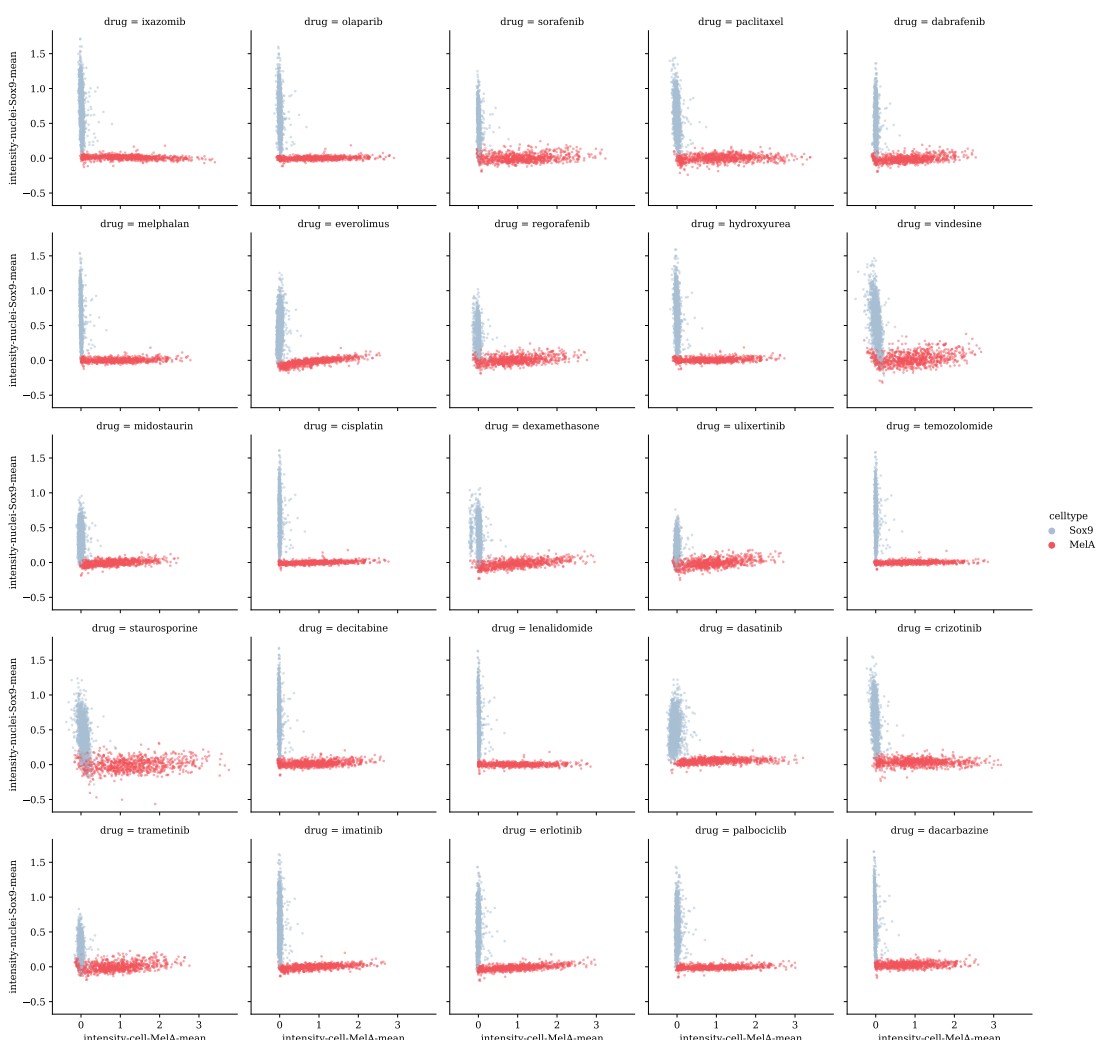

Figure 14: Classification of cells into cell types (MelA$^+$, Sox9$^+$) based on protein marker intensities of MelA and Sox9, for all drugs, at $t = 8h$ § C.2. Each tile represents one drug. MelA$^+$ cells colored in red, Sox9$^+$ in blue.

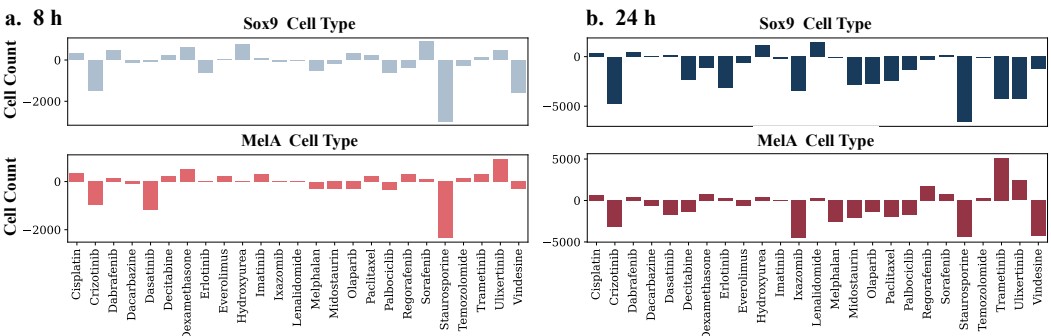

Figure 15: Drug treatment-induced change in cell counts in the two cell types compared to the cell count of the respective cell types in the control condition. **a.** Cell count change for cell types Sox9$^+$ (top) and MelA$^+$ (bottom) at $t = 8h$. **b.** Cell count change for cell types Sox9$^+$ (top) and MelA$^+$ (bottom) at $t = 24h$.

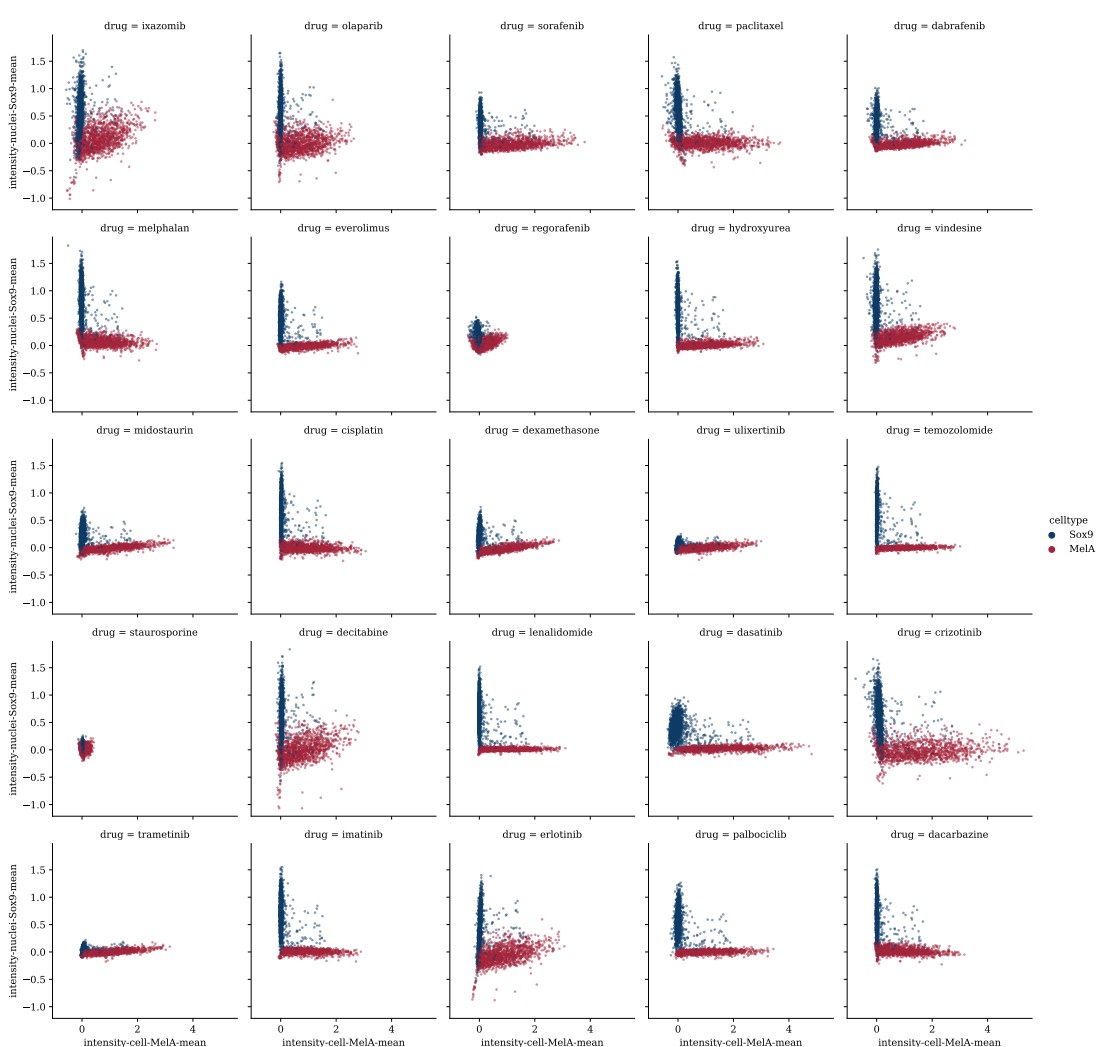

Figure 16: Classification of cells into cell types (MelA$^+$, Sox9$^+$) based on protein marker intensities of MelA and SOX9, for all drugs, at $t = 24h$ § C.2. MelA$^+$ cells colored in red, Sox9$^+$ in blue.

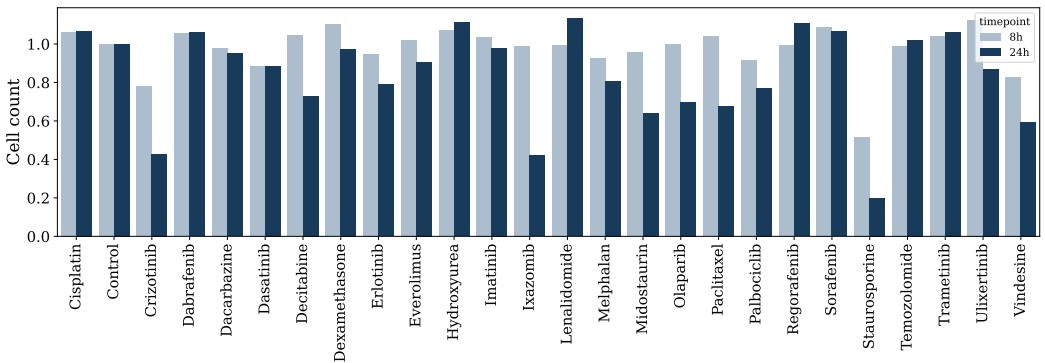

Figure 17: Observed cell counts of drug-treated cells normalized to control cell counts, per drug and time point. 8h treatment in light blue, 24h treatment in dark blue.

we then used to predict the cell type labels of the drug treated cells. The procedure was performed separately for the 8h- and 24h dataset. Results of the classification can be found in Figure 14 and Figure 16 respectively.

# D    EXPERIMENTAL DETAILS

NUBOT consists of several modules and its performance is compared against several baselines. In the following, we provide additional background on experimental details, including a description of the evaluation metrics and baselines considered, as well as further information on the parameterization and hyperparameter choices made for NUBOT.

## D.1    EVALUATION METRICS

We evaluate our model by analyzing the distributional similarity between the predicted and observed perturbed distribution. For this, we compute the kernel maximum mean discrepancy (MMD) (Gretton et al., 2012). We utilize the RBF kernel, and as is usually done, report the MMD as an average over several length scales, i.e., $2, 1, 0.5, 0.1, 0.01, 0.005$. To take the mass variation into consideration, we compute a weighted version of MMD, by weighting each predicted point by its associated normalized weight. Additionally, we compute the weighted Wasserstein distance between the predicted and observed perturbed cells (2).

## D.2    BASELINES

We compare NUBOT against several baselines, comprising a balanced OT-based method (Bunne et al., 2021, CELLOT) and an unbalanced OT-based method (Yang & Uhler, 2019, NUBOT), i.e., current state-of-the-art methods as well as ablations of our work. We further provide a comparison to the closed-form of entropy-regularized optimal transport on unbalanced Gaussians (Janati et al., 2020b). In the following, we briefly motivate and introduce each baseline.

**CELLOT.**    By introducing reweighting functions $\eta$ and $\zeta$, NUBOT recovers a balanced problem parameterized by dual potentials $f$ and $g$. An important ablation study to consider is thus to compare its performance to its balanced counterpart. Ignoring the fact that the original problem includes cell death and growth, and thus varying cell numbers, we apply ideas developed in Makkuva et al. (2020); Bunne et al. (2021) and learn a balanced OT problem via duals $f$ and $g$. These duals are parameterized by two ICNNs and optimized in objective (5) via an alternating min-max scheme.

**UBOT GAN.**    Using (6), Yang & Uhler (2019) propose to model mass variation in unbalanced OT via a relaxation of the marginals. Similar to Fan et al. (2021a), Yang & Uhler (2019) reformulate the constrained Monge problem (3) as a saddle point problem with Lagrange multiplier $h$ for the constraint $T_\sharp \mu = \nu$, i.e.,

$$\sup_h \inf_T \int_{\mathcal{X}} c(x, T(x))\mu(x)dx + \int_{\mathcal{X}} h(y)\left(\nu - T_\sharp \mu\right) dy$$
$$= \int_{\mathcal{X}} [c(x, T(x)) - h(T(x))]\mu(x)dx + \int_{\mathcal{X}} h(y)\nu(y)dy,$$

parameterizing $T$ and $h$ via neural networks. To allow mass to be created and destroyed, Yang & Uhler (2019) introduce scaling factor $\xi : \mathcal{X} \to \mathbb{R}^+$, allowing to scale mass of each source point $x_i$. The optimal solution then needs to balance the cost of mass and the cost of transport, potentially measured through different cost functions $c_1 : \mathcal{X} \times \mathcal{Y} \to \mathbb{R}^+$ (cost of mass transport) and $c_2 : \mathbb{R}^+ \to \mathbb{R}^+$ (cost of mass variation). Parameterizing the transport map $T_\theta$, the scaling factors $\xi_\phi$, and the penalty $h_\omega$ with neural networks, the resulting objective is

$$l(\theta, \phi, \omega) := \frac{1}{n} \sum_{i=0}^{n} [c_1(x_i, T_\theta(x_i))\xi_\phi(x_i) + c_2(\xi_\phi(x_i)) + \xi_\phi(x_i)h_\omega(T_\theta(x_i)) - \Psi^*(h_\omega(y_i))],$$

with $\Psi^*$ approximating the divergence term of the relaxed marginal constraints (see (6)), and is optimized via alternating gradient updates.

**GAUSSIAN APPROX.** Janati et al. (2020b) provide a closed-form solution of the entropy-regularized optimal transport problem on unbalanced Gaussians. They show that the unbalanced optimal transport plan is, minimizer of (7), is also a Gaussian distribution over $\mathbb{R}^d \times \mathbb{R}^d$.

In order to use this baseline on the single cell data, we first compute a Gaussian approximation of the control and treated cells separately in the original cell data space. Then, we compute the closed-form joint coupling, $\pi = \mathcal{N}(\boldsymbol{\mu}, \boldsymbol{\Sigma})$, where

$$\boldsymbol{\mu} = \left[ \begin{array}{c} \boldsymbol{\mu}_x \\ \boldsymbol{\mu}_y \end{array} \right] \text{ and } \boldsymbol{\Sigma} = \left[ \begin{array}{cc} \boldsymbol{\Sigma}_{xx} & \boldsymbol{\Sigma}_{xy} \\ \boldsymbol{\Sigma}_{yx} & \boldsymbol{\Sigma}_{yy} \end{array} \right]. \tag{16}$$

Given a test source sample $\mathbf{x}_t$, we then compute the conditional expectation of the transported target $\mathbf{y}$, i.e.,

$$\mathbb{E}_\pi \left[ \mathbf{y} | \mathbf{x} = \mathbf{x}_t \right] = \boldsymbol{\mu}_x + \boldsymbol{\Sigma}_{xy} \boldsymbol{\Sigma}_{yy}^{-1} \left( \mathbf{x}_t - \boldsymbol{\mu}_y \right). \tag{17}$$

**DISCRETE OT.** Additionally, we consider the entropy-regularized Wasserstein mapping returned by the Sinkhorn algorithm (Chizat et al., 2018a; Cuturi, 2013; Benamou et al., 2015) on finite sets. This algorithm does not return a parameterized solution, but rather a transport map between finite source and target sets. Thus, the solution is computed on the full dataset (inclduing train and test data) and cannot be considered in the out-of-sample setup. We have included a comparison anyway for completion.

**IDENTITY.** A trivial baseline is to compare the predictions to a map which does not model any perturbation effect. The IDENTITY baseline thus models an identity map and provides an *upper bound* on the overall performance, also considered in Bunne et al. (2021).

**OBSERVED.** In a similar fashion we might ask for a *lower bound* on the performance. As a ground truth matching is not available, we can construct a baseline for a comparison on a distributional level by comprising a different set of observed perturbed cells, which only vary from the true predictions up to experimental noise. The closer a method can approach the OBSERVED baseline, the more accurate it fits the perturbed cell population.

### D.3 HYPERPARAMETERS

We parameterize the duals $f$ and $g$ using ICNNs with 4 hidden layers, each of size 64, using ReLU as activation function between the layers. We choose the identity initialization scheme introduced by Bunne et al. (2022b) such that $\nabla g$ and $\nabla f$ resemble the identity function in the first training iteration. As suggested by Makkuva et al. (2020), we relax the convexity constraint on ICNN $g$ and instead penalize its negative weights $W_l^z$

$$R(\theta) = \lambda \sum_{W_l^z \in \theta} \|\max(-W_l^z, 0)\|_F^2.$$

The convexity constraint on ICNN $f$ is enforced after each update by setting the negative weights of all $W_l^z \in \theta_f$ to zero. Duals $g$ and $f$ are trained with an alternating min-max scheme where each model is trained at the same frequency. Further, both reweighting functions $\eta$ and $\zeta$ are represented by a multi-layer perceptron (MLP) with two hidden layers of size 64 for the single-cell and of size 32 for the synthetic dataset, with ReLU activation functions. The final output is further passed through a softplus activation function as we do not assume negative weights. For the unbalanced Sinkhorn algorithm, we choose an entropy regularization of $\varepsilon = 0.005$ and a marginal relaxation penalty of 0.05. We use both Adam for pairs $g$ and $f$ as well as $\eta$ and $\zeta$ with learning rate $10^{-4}$ and $10^{-3}$ as well as $\beta_1 = 0.5$ and $\beta_2 = 0.9$, respectively. We parameterize both baselines with networks of similar size and follow the implementation proposed by Yang & Uhler (2019) and Bunne et al. (2021).

## E   REPRODUCIBILITY

The code will be made public upon publication of this work.

