# OpenReview forum: "Neural Unbalanced Optimal Transport via Cycle-Consistent Semi-Couplings"
_ICLR.cc/2023/Conference — Submitted to ICLR 2023_

### Official Review · Reviewer_irGq · 2022-10-21

**Confidence:** 4
**Correctness:** 2
**Technical Novelty And Significance:** 2
**Empirical Novelty And Significance:** 3
**Recommendation:** 3

**Clarity, Quality, Novelty And Reproducibility:**

**Quality.** The proposed method uses the solutions of the discrete unbalanced OT problem to train networks to predict scaling factors. However, the solutions of discrete OT methods applied to batches are biased regarding the solutions of continuous / full batch problems. It is unclear to what extent their solutions and the learned networks match the actual solutions. Specifically, it is not evident how well the networks approximate the actual scaling factors, and, thus, how well the method approximates an actual OT map. In the proposed framework, I  think that it is necessary to show that such approximations are reasonable. The authors do not conduct a convincing experiment showing that the method calculates what is intended, i.e., finds an optimal transport solution. At the same time, the method is not supported by theoretical analysis.

*(a)* In the setup of the synthetic experiment, the authors consider the task of mapping unbalanced mixtures of Gaussians, for which, as the authors write, “a ground truth matching is available”. However, they did not include any comparison of the proposed method results with this “ground truth matching”. And how do you derive the ground truth OT maps between the mixtures?

*(b)* I recommend the authors to quantitatively evaluate the method using unbalanced Gaussians for which both the precise scaling factors and precise OT maps seem to be analytically known [1]. Thus, it is possible to rigorously quantitatively evaluate whether the proposed method actually recovers the precise OT map and scaling factors.

[1] Janati, H., Muzellec, B., Peyré, G., & Cuturi, M. (2020). Entropic optimal transport between unbalanced Gaussian measures has a closed form. Advances in neural information processing systems, 33, 10468-10479.

The main experiment section 4.3 of the paper focuses on the task of predicting single-cell responses to different treatments, which raises the problem of mapping unbalanced distributions of cells before and after the treatment. The authors provide many biological details regarding this problem which explain the choice of the task-specific metrics for comparison with other methods. Despite the fact that the method improves these metrics, I am concerned about this experiment because it is not transparent what happens inside the method (see my above comments). I am not familiar with the problem considered, so I wonder whether there are any non-OT based methods for it to compare with. The discussion of this aspect is limited.

Comparison with other unbalanced GAN OT method (Yang & Uhler, 2019) also might seem to be incomplete in some sense. Can the proposed method be applied to the other unbalanced OT tasks originally considered in (Yang & Uhler, 2019), e.g., in CelebA-Young/CelebA-Aged case? A comment about this is appreciated.

**Clarity.** In general, I think that some of the paper results are not well explained. From Section 4.1, subsection ‘Updating rescaling functions’, I do not completely understand why the measures re-scaled using obtained scaling factors are balanced? Section 4.3 is very hard to follow due to the large amount of biological terminology. In my view, this is a serious drawback of the paper because the experimental part may be hard to parse for ML practitioners without significant background in biology.

**Reproducibility.** The authors do not provide the code of their method. Thus, the reproducibility of their method is questionable. (However, some technical details are given in Appendix.).

**Minor remarks.**

(1) In the Equation (5) - squares of norms are missing, subscript n is undefined.

(2) In the Equation (7) - two subscripts m.

(3) In Section 3, subsection ‘Updating rescaling functions’ - typo in ‘our goal is to find w’.

(4) In Section 3, subsection ‘Transforming new samples’ - ‘mapped backed’ should be changed to ‘mapped back’.

(5) In Figure 3 - undefined parameters R and P.

(6) In section 4.1 typo in ‘(see Fig. 10, 12, as well as Fig. 10, 12).

**Strength And Weaknesses:**

**Strengths.** The paper proposes an optimal transport approach for an important unbalanced setup, which is not yet fully explored.

**Weaknesses.**
1. The proposed algorithm is based on the solutions of the discrete unbalanced OT problem. Thus, it is unclear to what extent the method approximates the actual underlying solution of the continuous problem. This might lead to bias in method solutions.
2. The proposed approach lacks comprehensive quantitative evaluation. It is not shown that it learns actual scaling factors and OT map between distributions.


**Summary Of The Paper:**

The paper proposes a method for finding optimal transport maps between unbalanced distributions, i.e., distributions with different total masses. The method is composed of two steps repeated iteratively. First, it seeks for scaling factors for source and target measures, such that the re-scaled measures are balanced. This problem is considered as a discrete unbalanced optimal transport problem and solved using the Sinkhorn algorithm. These factors are used to train neural networks to predict scaling factors for arbitrary points. At the second step,  the method finds OT maps between re-scaled measures using standard balanced OT formulation (Makkuva et al., 2020).

The method is tested on the synthetic problem of finding a map between unbalanced mixtures of Gaussians, and the real-world problem of testing cells' response to different drugs which may induce proliferation (quantitative growth) or death of cells, i.e., change in distribution mass.

**Summary Of The Review:**

The authors propose a method for finding an optimal transport map between unbalanced distributions. However, from the construction of the method it is unclear how well it approximates actual scaling factors and OT map between the distributions. The experimental section also does not clarify this issue. Due to this, I think that not enough evidence is given to convince the reader that the method computes what it is intended to do. Moreover, the entire focus of the method is a particular biological task, which is hard to follow for an unprepared reader. Because of all these issues, I am currently more on the negative side about the paper. If the authors solve these evaluation/clarity issues during the rebuttal phase, I will be happy to reconsider my assessment.

---

> ### Author Response · Authors · 2022-11-19
> **Response to Reviewer irGq (1/3)**
>
> Thank you for the thorough review and comments. We have corrected the spelling errors and unclear notation that you point out.
>
> > **It is unclear to what extent the method approximates the actual underlying solution of the continuous problem [...] solutions of discrete OT methods applied to batches are biased.**
>
> While this is certainly true (and applies to most applications of OT in large-scale settings), we disagree this is a fundamental issue with the method for two reasons. First, the bias from batching is most prominent in the coupling computation, but much less so for learning the $f, g$ potentials (which depend on single points and thus enjoy typical SGD approximation guarantees). But note that only the marginals of the OT couplings are used (Algorithm 1: l. 6 and 8), after a variance-reducing averaging across one of the two dimensions. Second, bias from batching can be mitigated by choosing sufficiently large batch sizes, i.e., it is not an irremediable issue. We have included an additional analysis of the effect of the batch size in Fig. 7 in Appendix §B.1.1.
>
> > **It is not evident how well the networks approximate the actual scaling factors, and, thus, how well the method approximates an actual OT map.**
>
> Indeed, it is challenging to assess the quality of the predicted scaling factors as (i) in most settings, there are no ground-truth scaling factors, and (ii) there are different formulations of unbalanced OT each designed to tackle particular types of problems, as explained in $A.1. In the cell experiments, we thus seek to evaluate NubOT through metrics that capture the distributional fit of the transported and reweighted sample to the observed target sample (Figure 3, 10), and through proxy metrics, such as per-group aggregate masses of the cell types (Figure 4), all of which suggest that the predicted scaling factors are capturing the unknown underlying scaling phenomenon appropriately.
>
> > **And how do you derive the ground truth OT maps between the mixtures?**
>
> There is no ground-truth solution for the mixture setting, we have clarified that. However, we know the proportions with which we have sampled the clusters, and compare the scaling factors to these ratios, assuming cluster correspondence. We have updated Figure 2 to contain the ground-truth cluster growth factors. We updated Table 1 in Appendix §B1 to provide further insights into how well NubOT captures the ground truth compared to ubOT GAN by Yang and Uhler, (2019). We have further added an additional analysis on the performance of standard ubOT to capture the ground truth (see Figure 8, 9). The results demonstrate that the choice of hyperparameters in standard ubOT is critical to find the true solution. Despite depending on similar parameters, the neural network-based and cycle-consistent architecture of NubOT induce a certain robustness towards certain ranges of hyperparameters (see Figure Fig. 7 in Appendix §B.1.1 for an additional analysis on the influence of hyperparameters).
>
> > **I recommend the authors to quantitatively evaluate the method using unbalanced Gaussians for which both the precise scaling factors and precise OT maps seem to be analytically known.**
>
> Thanks a lot for this suggestion! We have added the closed form of the entropy-regularized optimal transport problem on unbalanced Gaussians by Janati et al., (2021) as an additional baseline for all cell experiments. The baseline itself is described in §D.2. We rerun all experiments using this additional baseline (see Fig. 4 and 10). For completion and better assessment of the results, we have further included balanced discrete OT as an additional baseline despite being computed on the entire dataset (including train and test samples) and being unable to generalize out-of-sample.

---

> > ### Author Response · Authors · 2022-11-19
> > **Response to Reviewer irGq (2/3)**
> >
> > > **I wonder whether there are any non-OT-based methods for it to compare with.**
> >
> > This is indeed a fair remark. Bunne et al., (2022) demonstrated that OT-based methods for modeling perturbation responses outperform autoencoder-based methods such as scGen by Lotfollahi et al., (2019) and cAE based on Lopez et al., (2018). These baselines, however, do not incorporate cell birth and death processes. An additional non-OT-based method for modeling distribution shifts that incorporate population growth and shrinkage we are aware of is PRESCIENT by Yeo et al., (2021). Following Hashimoto et al., (2016), they parameterize population dynamics through learned potential functions of a stochastic differential equation but propose to extend that framework by incorporating cell growth and death. To achieve this, they “weight[...] each cell in the source population according to its expected number of descendants in the objective” using “*a priori* knowledge of cell proliferation”. In particular, they either “estimate [scaling factors] from the data by computing the number of descendants for each starting cell given lineage tracing data”, i.e., information not contained in most datasets, **or** compute “proliferation [...] from gene expression” data via “KEGG annotations of cell cycle and apoptosis genes which were also highly variable in the dataset to estimate the number of descendants a cell is expected to have”.
> > Our model, however, learns population growth and shrinkage *without* requiring prior knowledge. Instead, we infer cellular birth and death by simultaneously fitting the overall dynamics, and by doing so, is one of the only methods of this kind.
> >
> > In general, finding non-OT baselines for such cell experiments is very challenging. The lack of known alignments between the populations at different times precludes the use of most traditional methods, so unsurprisingly most prior works use OT-based methods for this problem (Hashimoto et al., (2016) use OT as a loss function). In case you are aware of further baselines, we are very keen on further comparing NubOT to alternative approaches.
> >
> > > **Comparison with other unbalanced GAN OT methods (Yang & Uhler, 2019) also might seem to be incomplete in some sense.**
> >
> > We provide a description of the ubOT GAN by Yang and Uhler (2019) in Appendix D.2. We compare against their method for all different scenarios and extensively screened hyperparameters to make it work. The hyperparameter setting associated to the best results on the 4i dataset is very close to the setting used on their cell experiments (see [config](https://github.com/uhlerlab/unbalanced_ot/blob/main/utils.py#L22)), i.e., $\lambda_0=0.01, \lambda_1=1, \lambda_2=1$, and for the gradient penalties $\lambda_G=10, \lambda_{G_2}=0.01$, using $\Psi^*$ as implemented [here](https://github.com/uhlerlab/unbalanced_ot/blob/main/main.py#L169). Could you thus elaborate in what sense you feel that the conducted comparison is incomplete?
> >
> > > **Can the proposed method be applied to the other unbalanced OT tasks originally considered in Yang & Uhler, (2019), e.g., in CelebA-Young/CelebA-Aged case? A comment about this is appreciated.**
> >
> > First, we would like to point out that Yang & Uhler, 2019 do not test their method directly on the CelebA-Young and CelebA-Aged dataset but in the latent space of an independently trained variational autoencoder (VAE) with unspecified dimensionality. The experiments on data of the zebrafish embryogenesis are conducted on a 50-dimensional PCA embedding. While this is not explicitly mentioned in the paper, this can be derived from the data provided in their [repository](https://github.com/uhlerlab/unbalanced_ot/tree/main/data). Thus, their experiments are conducted on problems of similar dimensionality as the settings considered in our paper.
> >
> > In general, our method can be applied to tasks considered in Yang & Uhler, 2019. Their code, however, does not provide us with enough details to reproduce the experiments. The chosen architecture of the VAE is not described in the paper, and neither hyperparameters nor code is provided (the [repository](https://github.com/uhlerlab/unbalanced_ot) only contains the cell experiments). From the experiments presented in their Figure 3, 4, and 5, it is not clear if the samples themselves were generated or only the matching between the unbalanced datasets evaluated. We tried our best but were not able to reproduce the setting from the information provided.
> >
> > Beyond class-imbalanced datasets, we are mainly interested in modeling birth and death processes, which is not the focus of the image experiments in Yang & Uhler, (2019). If you have further insights, we are happy to explore such directions.

---

> > > ### Author Response · Authors · 2022-11-19
> > > **Response to Reviewer irGq (3/3)**
> > >
> > > > **I do not completely understand why the measures re-scaled using obtained scaling factors are balanced?**
> > >
> > > Please see our general answer on this issue above.
> > >
> > > > **The authors do not provide the code of their method. Thus, the reproducibility of their method is questionable.**
> > >
> > > Although we will release our codebase upon publication, we believe we have provided all necessary details (algorithm, parameter choices, etc.) that would allow for replication of these results, and we are glad to see that the other reviewers seem to agree. If there are any particular aspects of the implementation that are not clear we will be more than happy to clarify them.

---

> > > > ### Comment · Reviewer_irGq · 2022-11-29
> > > > **Concern about the work**
> > > >
> > > > Dear authors, thanks for your answers and paper revision. The revised version is indeed more clear and transparent. However, I still have a very important concern, which, I think, is not properly addressed in the revision.
> > > >
> > > > In my review, I mentioned that “*it is unclear to what extent the solutions and the learned networks match the actual solutions*”. The authors included an additional analysis of the effect of the batch size on their method performance and stated that “*bias from batching can be mitigated by choosing sufficiently large batch sizes*”. In the newly added analysis, the authors consider **2D** mixtures of gaussians which can be well approximated by discrete OT methods using a large batch size. However, I doubt that using sufficiently large batch sizes tackling data of bigger dimensions is computationally feasible. It is an important issue since the authors state that their single-cell dataset “*consists of **48** features*” (Appendix C.2).
> > > >
> > > > In order to address this concern, I have proposed the authors to evaluate their method on unbalanced higher dimensional Gausians for which both the precise scaling factors and precise OT maps seem to be analytically known (Janati et al, 2021). The authors added the method on unbalanced Gaussians by Janati et al. (2021) as an additional baseline for their cell experiments (2D case). Indeed, it is a good baseline to compare with but this is not what I meant.
> > > >
> > > > My request is to compare the author’s method NubOT with the closed-form ground truth OT map and scaling factor on when distributions are themself unbalanced nD Gaussians. In this case, it is possible to assess the learned solutions using **explicit** metrics (e.g. $L_2$ norm between learned solutions and known ground truth mappings & scaling factors) as well as compare with ubOT GAN. In contrast to experiments conducted by the authors (which are mostly assessed with “**proxy**” metrics), the proposed experiment could help to clarify the actual performance of the proposed OT method. *Do I miss something, i.e., is there any reason to not conduct such an experiment?*
> > > >
> > > > This is my only major concern about this work and the reason why I tend to keep my score. It is still vague to which extent the discrete approximations of "continuous" unbalanced OT plans are appropriate in high dimensions. It would be great if the authors could explicitly detail this aspect of their method.

---

> > > > > ### Author Response · Authors · 2022-11-30
> > > > > **Follow-Up to Reviewer irGq**
> > > > >
> > > > > Thanks a lot for following up!
> > > > >
> > > > > > **I proposed to evaluate their method on unbalanced higher dimensional Gaussians for which both the precise scaling factors and precise OT maps seem to be analytically known (Janati et al, 2021).**
> > > > >
> > > > > In the [paper by Janati et al., (2021)](https://papers.nips.cc/paper/2020/file/766e428d1e232bbdd58664b41346196c-Paper.pdf), Theorem 3 proves that the unbalanced optimal transport **plan** is also an unbalanced Gaussian, i.e., $\pi = m_\pi \mathcal{N}(\mu, H)$. However, to the best of our understanding that paper does not provide a closed-form *map* between them. So, assuming that what was meant was to compare against the precise OT **plan/coupling**, that is something we can potentially do by marginalizing over the second argument and comparing the expectation of that marginal against our predicted mapped points. Concretely, we would compute the L2 distance between each of these 'ground truth barycentric projections' and the corresponding prediction of our method and then average these. We are happy to provide these results once we received confirmation this is what was meant.
> > > > >
> > > > > What is less clear is how to obtain 'precise scaling factors' from this joint Gaussian. If the reviewer has a specific approach in mind to do this, and it is something we can conceivably attempt in the remainder of the discussion period, we are more than happy to provide results using it too.
> > > > >
> > > > > > **I doubt that using sufficiently large batch sizes to tackle data of bigger dimensions is computationally feasible. It is an important issue since the authors state that their single-cell dataset consists of 48 features.**
> > > > >
> > > > > We disagree with your concern about computational feasibility. For example, a number of generative models producing high-quality images use optimal transport as subroutines in their training loop, i.e., [Genevay et al., (2018)](http://proceedings.mlr.press/v84/genevay18a/genevay18a.pdf), and thus are of the same computational complexity as our method. Even if we need to adapt the batch size for potential tasks of higher dimensions, recent implementations of (unbalanced) optimal transport methods scale very well up to $n=4096$, showing that solving such problems on much larger batch sizes would indeed be computationally feasible. For reference, see [this notebook](https://ott-jax.readthedocs.io/en/latest/notebooks/OTT_%26_POT.html) and [figure](https://ott-jax.readthedocs.io/en/latest/_images/notebooks_OTT_&_POT_16_0.png) for an analysis of sample size and runtime.

---

> > > > > > ### Comment · Reviewer_irGq · 2022-12-01
> > > > > > **More concerns**
> > > > > >
> > > > > > Dear authors,
> > > > > >
> > > > > > **1.** After reading the paper again, I understood that I do not completely realize which unbalanced OT problem are you trying to solve with your composite discrete-continuous algorithm.
> > > > > >
> > > > > > If this the entropic UBOT (7), then for it there is the exact formula (Theorem 3 in Janati et al.) for the Unbalanced OT plan between Gaussians (from which one may recover the scaling factors as well). This plan is non-deterministic while your algorithm learns a deterministic one (as you pointed in your answer). This raises additional questions what is actually being learned in practice.
> > > > > >
> > > > > > In Appendix A1, you write "This work proposes a method to parameterize an UBOT problem which does not necessarily match the typical Kantorovich entropy-regularized formulation." If the model you present does not learn the entropic OT, then what is the precise unbalanced mathematical problem which your algorithm aims to solve? I looked through the paper and did not find the direct answer.
> > > > > >
> > > > > > **2.** My question is not about the large computational complexity of discrete OT. My question is about how well the discrete OT can approximate continuous OT in high dimensions (e.g., 48) for computationally feasible batch sizes. I understand that there are approaches such as (Genevay et. al). Nevertheless, for me it is not obvious that the succesful application (Genevay et. al) of discrete entropic OT to GANs (which depend on many other things beside OT such as neural network architectures) necessarily means that it works well in you regression context where you need to recover the scaling factors.

---

> > > > > > > ### Author Response · Authors · 2022-12-02
> > > > > > > **Clarification of concerns of Reviewer irGq**
> > > > > > >
> > > > > > > Thank you for the clarification, and for the continued engagement. Despite the pushback, we greatly appreciate the effort/time you have put into digging deeper into our paper.
> > > > > > >
> > > > > > > > **What is the precise unbalanced mathematical problem which your algorithm aims to solve?**
> > > > > > >
> > > > > > > To put it more succinctly, our method does not solve a usual unbalanced OT problem directly, but instead solves a *rebalanced* one. Indeed, while our starting point is UBOT, as stated in the paper our ultimate objective is a balanced OT formulation, which allows us to leverage Brenier's theorem to get a Monge map. What this means is that we use usual UBOT (7) not as the main objective but rather to formulate the constraints of the optimization problem.
> > > > > > >
> > > > > > > More formally, the optimization problem we seek to solve is:
> > > > > > >
> > > > > > > $\inf_{f,g} \int_{\mathcal{X}}[f(\nabla g(x))-\langle x, \nabla g(x)\rangle] \eta(x) \mathrm{d} \mu(x)-\int f(y) \zeta(y) \mathrm{d} \nu(y)$
> > > > > > >
> > > > > > > subject to:
> > > > > > > 1. $D \bigl( (Proj_1)_\sharp [\gamma^{*}(T_\sharp \mu, \nu)] , T_\sharp(\tilde{\mu}) \bigr) =  0$
> > > > > > > 2. $D\bigl((Proj_1)_\sharp [\gamma^*(S_\sharp \nu, \mu)] , S_\sharp(\tilde{\nu}) \bigr) =0 $
> > > > > > > 3. $T = \nabla f, S = \nabla g, \tilde{\mu} = \eta \cdot \mu,  \tilde{\nu} = \zeta \cdot \nu $.
> > > > > > >
> > > > > > > In practice, we turn the first two constraints into empirical (discrete) soft penalties as:
> > > > > > >
> > > > > > > $\min_{\eta, \zeta}  \| \eta(\mathbf{x}) - \Gamma^*(T_\sharp \mu_\mathbf{x}, \nu_\mathbf{y})\mathbf{1}\|^2 + \| \zeta(\mathbf{y}) - \Gamma^*(S_\sharp \nu_\mathbf{y}, \mu_\mathbf{x})\mathbf{1}\|^2$
> > > > > > >
> > > > > > > where now the $\Gamma^*$'s are the solutions of the discrete UBOT problems (7), and $\mu_\mathbf{x}, \nu_\mathbf{y}$ indicate the empirical distributions supported on samples $\mathbf{x}$ and $\mathbf{y}$.
> > > > > > >
> > > > > > > > **My question is not about the large computational complexity of discrete OT. My question is about how well the discrete OT can approximate continuous OT in high dimensions (e.g., 48) for computationally feasible batch sizes. I understand that there are approaches such as (Genevay et. al). Nevertheless, for me it is not obvious that the succesful application (Genevay et. al) of discrete entropic OT to GANs (which depend on many other things such architectures) necessarily means that it works well in your regression context where you need to recover the scaling factors.**
> > > > > > >
> > > > > > > Again, thank you for reformulating the question. We agree that the abundant theoretical/empirical evidence of asymptotic approximation of the classic discrete OT problem does not immediately translate to this setting. Indeed, while each of the components can be estimated faithfully with reasonable sample-complexity rates (including the scaling factors, which are ultimately estimated by approximating the marginals of well-behaved entropic UBOT problems) , their combination might not enjoy the same guarantees, particularly for the scaling factors. This is an important theoretical question for future work, which most likely deserves its own entire paper, thus falling outside the scope of this one.

---

> > > > > > > > ### Author Response · Authors · 2022-12-09
> > > > > > > > **Follow-Up to Reviewer irGq**
> > > > > > > >
> > > > > > > > Dear Reviewer,
> > > > > > > >
> > > > > > > > Thank you very much for taking the time to review our paper. Unfortunately, the time window to interact with reviewers is ending soon, i.e., on **December 12th**. We hope our comments have addressed your concerns. Is there any more information we can provide?
> > > > > > > > If not, we would greatly appreciate if you could update your score given the changes made to the paper.
> > > > > > > >
> > > > > > > > Thanks again,
> > > > > > > >
> > > > > > > > the authors

---

### Official Review · Reviewer_rxRW · 2022-10-25

**Confidence:** 4
**Correctness:** 3
**Technical Novelty And Significance:** 3
**Empirical Novelty And Significance:** 2
**Recommendation:** 6

**Clarity, Quality, Novelty And Reproducibility:**

- The paper is well-written and easy to follow.
- Code is not available.


**Strength And Weaknesses:**

### Strength ###
- Formulation UBOT with a neural approach based on semi-couplings proxy measures. The Kantorovich potentials are learned with ICNNs.
- Extensive numerical study on synthetic and real data with an accent on cell-biology applications: prediction of cell proliferation and death, and the perturbation responses on the level of single cells.

### Weaknesses ###
- Steps 5 and 7 in Algorithm 1 deal with calculating an unbalanced Sinkhorn: it is a little be confused since the ultimate goal of this paper is to tackle UBOT.
- The paper lacks comparisons with the SOTA approaches of UBOT: NUBOT is only compared to ubOT GAN? One can compare it to a vanilla generalized Sinkhorn algorithm (Chizat et al. 2018).
- What is the computational complexity of NUBOT. I think this point should be clarified or at least the authors can give some insight bout it. Steps 5 and 7 in Algorithm 1 might be the most expensive ones.
- In Algorithm 1, it is not clear to me the update of $\eta$ and $\zeta$.


**Summary Of The Paper:**

Optimal transport distances (a.k.a. Wasserstein distance) have recently drawn ample attention in statistics and machine learning communities as powerful discrepancy measures for probability distributions. The standard formulation of OT assumes equality of mass to be transported from the source to the target, which is referred as to the conversation mass constraint. In other applications, this constraint is not filled; hence application of standard OT is not done directly. Existing lines of research investigate this problem by handling the non-equality of mass transportation. They consist of unbalanced Optimal transport (UBOT). These approaches are given by the sum of an OT cost and two constraints involving the non-equality of mass transportation. The second term is often written using divergence, for example, a Kullback-Leibler divergence, etc.

This paper proposes NUBOT, a neural unbalanced OT that relies on the formalism of semi-couplings to account for creation and destruction of mass. NUBOT takes advantage of the neural optimal transport, where it was shown that the Monge plan can be learned via the Kantorovich potentials of the dual problem, which can be parametrized via input convex neural networks (ICNNs). Towards this end, the authors suggest learning a proxy measures $\tilde{\mu} = \eta \cdot \mu$ and  $\tilde{\nu} = \zeta \cdot \nu$ to rescale the mass. The functions $\eta, \zeta$ are also parametrized via NN. An algorithm of NUBOT is given with an extensive numerical study with an accent on cell-biology applications.

**Summary Of The Review:**



### Typos ###
- Page 3: Eq. (3): there is no dependency on $\varepsilon$.
- Page 3: Eq. (5): OT$_$n. I don't understand this dependence on n.
- Page 4: The notation of the set of semi-coupling between measures $Gamma(\mu, \nu)$ is the same as the set of couplings in the nonnegative Radom measures (page 2).
- Page 4:
$\Gamma \gets \text{UBOT}(\boldsymbol{u}, T(\boldsymbol{x}_i), \boldsymbol{v}, \boldsymbol{y}_j)$ --> $\Gamma \gets \arg\min\text{UBOT}(\boldsymbol{e}\odot\boldsymbol{u}, T(\boldsymbol{x}_i), \boldsymbol{v}, \boldsymbol{y}_j)$
- Page 4:
$\Gamma \gets \text{UBOT}(\boldsymbol{v}, S(\boldsymbol{y}_j), \boldsymbol{u}, \boldsymbol{x}_i)$ --> $\Gamma \gets \text{UBOT}(\boldsymbol{z}\odot\boldsymbol{v}, S(\boldsymbol{y}_j), \boldsymbol{u}, \boldsymbol{x}_i)$
- Page 6: "intervention respnses" --> "intervention responses"

---

> ### Author Response · Authors · 2022-11-19
> **Response to Reviewer rxRW**
>
> Thanks for your thorough review, and for pointing out spelling errors and unclear notation. We have corrected this! In addition, we have changed the notation for the set of semi-couplings to $\Gamma_{||}$ in order to distinguish it from that of proper couplings.
>
> > **Steps 5 and 7 in Algorithm 1 deal with calculating an unbalanced Sinkhorn: it is a little be confused since the ultimate goal of this paper is to tackle UBOT.**
>
> We do not fully understand the concern here, but we want to clarify that the `unbalanced.sinkhorn` function solves an entropy-regularized version of the unbalanced discrete OT problem (UBOT). But it does not provide an OT *map* between the samples, and thus cannot be used to map out-of-sample. Thus while our goal is to tackle UBOT too, we seek a parametric solution that the vanilla UBOT algorithms do not provide. Instead, we use `unbalanced.sinkhorn` to handle the 're-balancing' aspect of UBOT, and neural OT to learn the mapping. We hope this clarifies things, but if it does not answer the concern we're happy to elaborate more.
>
> > **The paper lacks comparisons with the SOTA approaches of UBOT [...] One can compare it to a vanilla generalized Sinkhorn algorithm (Chizat et al., 2018).**
>
> The method of Chizat et al., (2018) is not applicable to the out-of-sample setup. While we can compute a mapping between two provided point clouds, it does not allow us to make predictions on unseen cells as no parameterization is provided. Could you elaborate which approach w.r.t. “vanilla generalized Sinkhorn algorithm (Chizat et al., 2018)” you have in mind?
>
> To provide some insights into the benefits of NubOT, we include discrete OT as an additional baseline, which, however, is computed on both train and test data and thus not directly comparable (see Fig. 3 and 10). As mentioned in the Response to All Reviewers, we compared results obtained by both UBOT and NubOT on different settings of the synthetic data example where the scaling factors are known. In addition, we compared hyperparameter sensitivities of both NubOT (§B1.1, Fig. 7) and UBOT (§B1.2, Fig. 8 and 9). The results demonstrate that NubOT is more robust to different hyperparameters and more reliably returns the correct scaling factors compared to UBOT.
>
> > **What is the computational complexity of NUBOT [...] Steps 5 and 7 in Algorithm 1 might be the most expensive ones.**
>
> Indeed, the complexity of the Algorithm is dominated by the computation of `unbalanced.sinkhorn`, which is O(nmT), where $T$ is the number of iterations of Sinkohrn. As a reference, for the balanced problem, an $\epsilon$-accurate solution of the OT problem can be found via Sinkhorn in $\tilde{O}(nm\epsilon^{-3} )$ where $\tilde{O}$ hides poly-log factors (Altschuler et al. 2017). Lines 6 and 8 are O(nm), line 9 is $O(n d B_g + m F_f)$, where $B_g$ is the cost of computing the gradient of $g$ (typically proportional to the number of NN parameters), and $ F_f$ is the cost of a forward pass on $f$. Lines 10 and 11 are just O(n) and O(m). Naturally, the empirical runtime will depend on various factors, including hidden constants. In practice, we observe that most of the computation is spent in computing  `unbalanced.sinkhorn`.
>
> > **In Algorithm 1, it is not clear to me the update of $\eta$ and $\zeta$.**
>
> In Algorithm 1, $e$ and $z$ are the empirical reweighting vectors needed to satisfy Eq 10 and its analogous counterpart for z. However, these are only valid for the current batch. In order to be able to predict reweighting for out-of-sample points, we learn to approximate $e$ and $z$ via neural networks $\eta$ and $\zeta$ which are simply trained as regressors on $e$ and $z$ via a Mean Squared Error loss.

---

> > ### Comment · Reviewer_rxRW · 2022-11-24
> > **Thanks for the detailed rebuttal**
> >
> > I thank the authors for their detailed responses, which suitably addressed my questions/comments. I also note that there is a significant enhancement of the NubOT approach. Given this, I am raising my score from 5 to 6.

---

### Official Review · Reviewer_zLtd · 2022-11-01

**Confidence:** 2
**Correctness:** 4
**Technical Novelty And Significance:** 3
**Empirical Novelty And Significance:** 3
**Recommendation:** 6

**Clarity, Quality, Novelty And Reproducibility:**

Clarity:

The paper is well-organized and clearly written.

Quality&Novelty:

The paper is of good quality for me and the technical contributions are clearly stated. However, I am an emergent reviewer and not an expert on OT. I'm not sure whether the literature is reviewed properly and whether the evaluation setup is fine. Therefore, I cannot justify its quality and novelty confidently.

Reproducibility:

It presents the experimental details including the data preprocessing, network architectures, and hyperparameter tuning process. It seems reproducible.

**Strength And Weaknesses:**

**Strength**

1. The paper is well-organized and clearly written.

2. The empirical results (both qualitative and quantitative ones) seem pretty good.

**Question**

I have several questions here.

1. Why use MMD instead of the OT distance as the evaluation metric in Fig. 4? It is more natural to evaluate the objective on the validation dataset for machine learning algorithms.

2. The method presented in Section 3 seems complex to me. It would be better to discuss the necessity for each part in depth and present the empirical contribution of each part in the experiments.

**Summary Of The Paper:**

First, I am an emergent reviewer and not an expert on OT. Therefore, I'm not sure whether the literature is reviewed properly and whether the evaluation setup is fine.

This work presents a formulation of the unbalanced optimal transport problem. To solve the unbalance problem, it relies on the formalism of semi-couplings to account for the creation and destruction of mass.

It further presents a cycle-consistent training procedure upon neural OT to estimate such semi-couplings and generalize out-of-sample, named NUBOT.

The method is evaluated on a synthetic dataset and the single-cell perturbation task. It is compared to CellOT (a balanced neural OT method) and UBOT GAN (an imbalanced OT approach).

According to the reweighted MMD (and other metrics), NUBOT outperforms the baselines on both datasets. Moreover, on the challenging single-cell perturbation task, NUBOT is able to successfully predict perturbed cell states, while explicitly modeling death and proliferation.

**Summary Of The Review:**

This is a solid paper on addressing the unbalanced OT problem and I tend to accept it now.

---

> ### Author Response · Authors · 2022-11-19
> **Response to Reviewer zLtd**
>
> Thanks a lot for your review and the provided feedback!
>
> > **I am an emergent reviewer and not an expert on OT. Therefore, I'm not sure whether the literature is reviewed properly and whether the evaluation setup is fine.**
>
> Beyond reviewing related concepts in Section 2 (Background) and introducing the notion of semi-couplings, we provide an extensive literature review in Section A on unbalanced OT (§A.1), cycle-consistent learning (§A.2), and convex neural architectures (§A.3). Further, we compared our literature to a [review](https://arxiv.org/pdf/2211.08775.pdf) article published coincidently throughout the course of this rebuttal and added the few additional references to §A.1 in the updated manuscript.
>
> **Why use MMD instead of the OT distance as the evaluation metric in Fig. 4? It is more natural to evaluate the objective on the validation dataset for machine learning algorithms.**
>
> Thanks for this remark! We have added entropy-regularized Wasserstein additional evaluation criterion. We updated Figure 3 (with additional baselines) showing MMD and Figure 10 displays the Wasserstein metric.
>
> > **The method presented in Section 3 seems complex to me [...] discuss the necessity for each part in depth and present the empirical contribution of each part in the experiments..**
>
> The actual method can be thought of as only having two parts: a rescaling-finding component (to account for mass creation/destruction) and a mapping estimation. Both are necessary: the former without the latter would have no basis on which to estimate the necessary rescaling, while the latter without the former would be a usual OT, incapable of modeling mass creation/destruction. Thus, an ablation study is not possible in this case, as removing either of these would result in a fundamentally different method.
>
> > **On Novelty.**
>
> In your review, you state that you *cannot justify its quality and novelty confidently*, but then score the paper 2/3 for technical/empirical novelty.  Yet this assessment is not accompanied by concrete weakness or concerns on the novelty. We would appreciate it if you can provide us with more actionable concerns that we could try to address.

---

> > ### Comment · Reviewer_zLtd · 2022-11-21
> > **Acknowledgement to the rebuttal**
> >
> > Thanks for the rebuttal, which addresses my concerns. I change my score on technical novelty.

---

### Official Review · Reviewer_CvpT · 2022-11-03

**Confidence:** 5
**Correctness:** 3
**Technical Novelty And Significance:** 2
**Empirical Novelty And Significance:** 3
**Recommendation:** 5

**Clarity, Quality, Novelty And Reproducibility:**

The experimental sections are well motived and clear. However, the presentation of the algorithm, and how it relates to the UBOT problem could be made more clear.

The novelty is modest, combining work on ICNN based OT and unbalanced transport via rescaling as in Yang and Uhler 2019 and Tong et al. 2020.

The experiments are largely reproducible.

**Strength And Weaknesses:**

Strengths:

- Important extension to the CellOT formulation which accounts for relative size changes in cell populations, which is extremely important in heterogenous cell systems.
- Well grounded theoretical formulation of unbalanced OT with semi-couplings.
- Good dataset choice for the evaluation of the unbalanced transport with a ground truth measurement of the number of cells at a population level before and after treatment.

Weaknesses:

- The connection between Algorithm 1 and the presented theorems is not entirely clear. From Algorithm 1 I would have expected the transformation for new samples to be $(x, u) \to (\nabla g(x), \eta(x) \cdot u)$. Could the authors clear up what is correct, algorithm 1 or figure 1?
- To me “forecasting heterogenous responses of multiple cancer cell lines to various drugs” would imply extrapolation to unseen drugs or cell line or at least combinations, instead of the forecasting the response of an unobserved cell in a seen drug and cancer cell line as is done here. Especially considering a separate model was trained for each treatment. Would it be possible to tone down the language on applications? In particular the "This is an unprecedented achievement in the field of single-cell biology" line? Modelling in distribution responses with proliferation has been tackled in prior work for time series, applying it to a seen perturbation response is in my view a relatively straight forward extension.
- The application is extremely specific, it is unclear how interesting it is to a wider ICLR audience. It would be useful to discuss other applications for unbalanced OT such as domain alignment, transfer learning etc.

Comments / Questions:

- In the “Updating Mappings” paragraph: “Since these are guaranteed to be balanced due to the argument above”. Could the authors clarify this please? Why is this guaranteed? How is guaranteed that $\int \tilde{\mu} = \int \tilde{\nu}$?
- There is something that is not lining up for me between the optimization in algorithm 1 and how samples are transformed. To me it seems like you would want to weight $\hat{y}$ by
- With a batchsize less than the entire data this is similar to a mini-batch OT problem. The unbalanced minibatch OT problem is in some senses creates a similar (full) map to the (full) balanced OT problem (Fatras et al. ICML 2021). Further, it doesn’t have the same weights as the regular non-batched problem. Does batch size have any affect on your results? I think this should at least be mentioned as not solving the same problem as UBOT.
- I’m curious how the learned weights compares to a discrete UBOT algorithm. I understand that there are advantages of learning a continuous map, however, I would still like to see a comparison to understand how accurately the proposed NubOT algorithm solves UBOT.
- I don’t believe that there are any constraints on the total sum of the transported measure $Z = \int_{(x,u)} \eta(x) \cdot u \cdot \zeta( \nabla(g(x))^{-1})$. The weighted MMD is invariant to Z, thus there is no evaluation whether the total measure is accurate. Is this total mass at all important to your application?
- Is there (albeit small) data leakage from data normalization in the single cell data to the 75% level?
- How was the relaxation penalty chosen, and do the authors expect it to be tuned on new datasets? What was the batch size chosen?

Minor Comments:

- Line 7 of Algorithm 1 should have an $x$ instead of $y$. Line 1 might want to be batches instead of epochs?
- I would note that $L^1$ norm unbalanced OT is not exactly the same as partial OT (As stated in A.1). Partial OT is slightly different. Instead this $L^1$ norm is sometimes “Robust optimal transport” (Mukherjee et al. ICML 2020).
- Could the true weights in Table 1 be depicted in Figure 2? Its difficult to understand that these are the correct weights without scrolling to Table 1.
- What is the weighted kernel MMD? What is the kernel? Gaussian with what bandwidth?

**Summary Of The Paper:**

This paper extends CellOT to the case of unbalanced optimal transport, which is essential to modelling most cell systems. This extension is accomplished through the formalism of semi-couplings, learning reweighted distributions which are then transported with balanced transport, in this case using the ICNN framework. A novel algorithm is presented to solve this problem efficiently through gradient descent. This algorithm is evaluated on a toy dataset and a single-cell drug perturbation dataset captured with 4i imaging. NubOT fits the data better in that it is able to predict cell states for unseen cells (but in seen conditions) better than comparable methods.

**Summary Of The Review:**

Overall a useful improvement concept with a clear and specific application. There are a few clarifications on the algorithm and the experiments. If these things are cleared up I have no problem raising my score.

---

> ### Author Response · Authors · 2022-11-19
> **Response to Reviewer CvpT (1/2)**
>
> Thank you for your elaborate review and the on point remarks!
>
> > **I would have expected the transformation for new samples to be (x,u)→(∇g(x),η(x)⋅u). Could the authors clear up what is correct, algorithm 1 or figure 1?**
>
> The transformation of a point $x$ with mass $u$ from the source $\mu$ to the target $\nu$ is $(x, u) \mapsto (\nabla g (x), \eta(x) \cdot u \cdot \zeta(\nabla g (x))^{-1})$. More specifically, $\nabla g(x)$ transports $x$ in the feature space. $\eta(x) \cdot u$ rescales the mass to the proxy measure $\tilde{\nu}$. As $\zeta(\cdot)$ rescales the mass of some point on the way backward, i.e. from $\nu$ to $\hat{\nu}$, we have to apply the inverse of this rescaling, that is $\zeta(\nabla g (x))^{-1})$, on the way forward, i.e. from $\hat{\nu}$ to $\nu$. Both Algorithm 1 and Figure 1 are correct.
>
> > **To me “forecasting heterogeneous responses of multiple cancer cell lines to various drugs” would imply extrapolation to unseen drugs or cell lines or at least combinations, instead of forecasting the response of an unobserved cell in a seen drug and cancer cell line.**
>
> This is indeed an interesting challenge. Extrapolating to unseen drugs and in particular combination therapies is extremely challenging and out of the scope of this work. The considered dataset further does not allow us to predict the outcome of combinations of drugs. In this paper, we concentrate on developing a robust model for cell birth and death, crucial when modeling and understanding a patient’s treatment response and we are the first to robustly achieve this.
>
> > **It would be useful to discuss other applications for unbalanced OT.**
>
> Unbalanced OT arises in any setting where one seeks a notion of distance or correspondence across datasets/populations *and* (i) the populations are not normalized in the same way and/or (ii) one does not want to rigidly enforce full correspondence between the datasets (e.g., to minimize the effect of outliers or because there is reason to believe some data points don't occur in both datasets). Some concrete applications in machine learning where UBOT is useful:
> - Domain adaptation / transfer learning, in particular for classification datasets that have non-uniform and unbalanced class distributions (e.g., class 1 over sampled in dataset A but oversampled in dataset B). In such cases, a usual OT formulation would necessarily force mappings across different classes, while the UBOT can avoid this by upscaling/downscaling samples from certain classes as necessary.
> - Color and shape matching: distributions might have arbitrary masses, or often a normalized measure doesn't even make sense.
> - Positive unlabeled classification (see an example from Chapel, Alaya, and Gasso, 2020).
>
> > **Responses to Questions**
>
> 1. **Why are the proxy measures guaranteed to be balanced?**
>
> Please see the general comment on this issue above.
>
> 2. **Does batch size have any affect on your results?**
>
> We run an additional evaluation testing the influence of the batch size on the model performance. Due to computational constraints and the number of additional experiments conducted throughout the course of this rebuttal, we were only able to rerun the experiments for four different batch sizes, i.e., 250, 300, 350, and 400. You can find the evaluation in the updated manuscript, Figure 7. The weights obtained from NubOT are consistent across all different batch sizes, showing that the batch size does not have a significant effect on our results.
>
> 3. **I think this should at least be mentioned as not solving the same problem as UBOT. How do the learned weights compare to a discrete UBOT algorithm?**
>
> This is a valid point, and we have added a paragraph in §A1 discussing this in the updated manuscript. Our main point is that UBOT is better thought of as a *family* of problems, rather than a single specific problem. This can be seen from the related work we cite here and in the paper, which poses UBOT in various different ways. In view of this, we would clarify your statement here to say that although we **propose a method to solve the UBOT problem**, it is indeed not the same formulation as the more typical Kantorovich entropy-regularized version of it. We particularly model problems in which different parts of the distribution have different changes in size. Further, our formulation allows for out-of-sample instance mapping.
>
> 4. **Is this total mass at all important to your application?**
>
> While not *per se* important to our application, the total mass provides useful insights for qualitative interpretation of the results. This can be seen in the evaluation we conducted in Section 4.2. Figure 4 shows a high correlation between observed cell counts of the two cell types and the sum of the predicted weights of the respective cell types after 8h of treatment for all drugs. We rewrote parts of §4 and hope this resolves the confusion.

---

> > ### Author Response · Authors · 2022-11-19
> > **Response to Reviewer CvpT (2/2)**
> >
> > 5. **Is there (albeit small) data leakage from data normalization in the single cell data to the 75% level?**
> >
> > We reprocessed the dataset: As before, we normalized the extracted intensity and morphological features by dividing each feature by its 75th percentile, computed on the control cells of the *training set*. The results do not vary and the performance is not affected.
> >
> > 6. **How was the relaxation penalty chosen, and do the authors expect it to be tuned on new datasets? What was the batch size chosen?**
> >
> > Both for the synthetic experiments and for the cell data, we chose the entropy regularization parameter $\epsilon = 0.005$ and the relaxation penalty $\tau_1 = \tau_2 = 0.05$, following the example provided [here](https://pythonot.github.io/auto_examples/unbalanced-partial/plot_unbalanced_OT.html#sphx-glr-auto-examples-unbalanced-partial-plot-unbalanced-ot-py) and being approriate in terms of runtimes. Since we normalize the obtained weights (see Algorithm lines 6, 8), we observed that our solution is fairly robust w.r.t. these parameters. We provide additional results on the effect of different hyperparameters including the batch size in Figure 7. For the cell data, we use a batch size of 256.
> >
> > > **The novelty is modest, combining work on ICNN-based OT and unbalanced transport via rescaling as in Yang and Uhler (2019) and Tong et al., (2020).**
> >
> > We respectfully disagree: These two works tackle the problem very differently. Yang and Uhler (2019) require careful tuning of the balance between two objectives, and Tong et al., (2020) do not actually model cell growth directly. See detailed comparisons below:
> >
> > Tong et al., (2020) indeed comment on the importance of modeling cell birth and death. Their *growth rate regularization* module, however, is trained independently using a regression network matching *pre-computed* growth rates (see [implementation](https://github.com/KrishnaswamyLab/TrajectoryNet/blob/master/TrajectoryNet/train_growth.py#L63)). In fact, all results on modeling cellular dynamics upon perturbation benchmarked on different datasets do not contain the growth model at all (denoted as $G$, see their Table 2 and 3). An analysis of capturing birth/death processes is only provided as a separate experiment (see their Fig. 7), in which the dynamics are learned independently (a much easier task). Tong et al., (2020) thus do not provide a general and robust implementation to parameterize the unbalanced optimal transport map. On the contrary, they clearly state that while “this is a necessary extension [it] is by no means a focus of [their] work” (§ 4.2, p. 6) and highlight “learning the growth term together with the dynamics”, i.e., the problem we tackle in NubOT, as one of the future directions (§6, p. 9). We further would like to emphasize that Tong et al., (2020) use a continuous normalizing flow for which one needs to “access to the density function of the source distribution. Since this is not accessible for an empirical distribution, [they] use an additional Gaussian at $t_0$”. Thus, their setting does not allow to predict perturbed states for *unseen* cells *at test time*, i.e., the setting we consider in this work (see [implementation](https://github.com/KrishnaswamyLab/TrajectoryNet/blob/master/TrajectoryNet/eval.py#L50)).
> >
> > Yang and Uhler (2019), on the other hand, propose parameterizing the unbalanced OT problem through the primal and by rescaling the weights. As introduced in § D.2, using this approach results in finding an optimal solution that needs to balance the cost of transport (i.e., a penalty of matching different samples) and cost of mass (i.e., a penalty of rescaling mass of a sample). For different datasets, finding a robust balance between both cost functions is challenging. Further, the chosen GAN objective and parameterizing $T$ of the primal problem results in unstable training.
> >
> > In this work, we build upon previous successes of ICNN-based optimal transport and propose an extension to model the unbalanced optimal transport scheme. Our method is theoretically supported through the notion of semi-couplings, a concept not previously utilized in neural parameterizations of OT. The chosen architecture and unbalanced scheme are novel and demonstrate strong performance and robustness on different datasets.
> >
> > > **Additional Remarks.**
> >
> > Thanks for pointing out the spelling errors and formatting suggestions. We corrected them and added additional references, as well as details on the MMD computation.

---

> > > ### Author Response · Authors · 2022-12-09
> > > **Follow-Up to Reviewer CvpT**
> > >
> > > Dear Reviewer,
> > >
> > > We are thankful for the time and effort you have put into reviewing our paper. The deadline to interact with reviewers is ending on **December 12th**. Could you please confirm that you have had a chance to look at the new elements we have provided in the rebuttal?
> > >
> > > We hope that the new experiments, explanations, and revisions that we have made to the paper have addressed your concerns. Please do not hesitate to ask more questions should you have any! We are looking forward to receiving your response.
> > >
> > > Best,
> > > the Authors

---

> > > > ### Comment · Reviewer_CvpT · 2022-12-12
> > > > **Thank you for your rebuttal and sorry for the late response**
> > > >
> > > > Thank you for your substantial effort on this rebuttal and for the new results. I believe the paper is substantially improved with the new comparisons to a standard discrete unbalanced OT. However, there are still some outstanding questions.
> > > >
> > > > In the general comments it is claimed that
> > > > > *NubOT is more robust to different hyperparameter choices* than recovering the weights from the generalized Sinkhorn algorithm by Chizat et al., (2018) directly.
> > > >
> > > > Citing new results in Figure 7. I don't believe this claim is substantiated by Figure 7. To me it looks like an underfitting problem with NubOT. I would have hoped to see that $\tau$ controls the "unbalancedness" similar to in standard unbalanced OT. The fact that the weights are the same for NubOT for the tested $\tau$ values suggests to me that the current network is unable to learn the balanced problem.
> > > >
> > > > As to novelty / relation to other unbalanced OT works, I do think this formulation is novel, but I believe there needs to be more validation to understand where it fits relative to other methods of tackling the unbalanced OT problem.
> > > >
> > > > For example, I'm unclear of how to control how "unbalancedness" of the problem. A combination of the robustness to $\tau$ and the batch size suggest to me that this is not well understood at this time. This is complicated by the introduction of an unbalanced neural network, transport ICNN, mini batch OT, and the semi-couplings, all of which can have an effect on the solution.
> > > >
> > > > It might be helpful to add these elements one at a time by starting with a standard balanced OT map in the middle (instead of the ICNN), and full batch unbalanced OT (instead of mini batch + learned weights) for the semi-couplings.
> > > > * Add in the ICNN
> > > > * Add in the learned semi-coupling weights
> > > > * Add in mini batch unbalanced OT
> > > >
> > > > With the added fixes of the data leakage and new parameter explorations it is clear to me that NubOT has a small but robust advantage over CellOT on this problem domain, but its still unclear exactly why this happens and if it can be reproduced in new settings.

---

> > > > > ### Author Response · Authors · 2022-12-13
> > > > > **Follow-Up**
> > > > >
> > > > > Dear Reviewer,
> > > > >
> > > > > Thank you for your response!
> > > > >
> > > > > > **I would have hoped to see that controls the "unbalancedness" similar to in standard unbalanced OT.**
> > > > >
> > > > > We have added a paragraph in §A1 in the paper that addresses the relation of NubOT to the entropy-regularized unbalanced OT (standard UBOT), where we state that we do not seek to solve standard UBOT.
> > > > >
> > > > > In the standard UBOT algorithm, $\tau$ controls the `unbalancedness' by specifying the penalty on the extent of marginal constraint violation. Allowing more unbalancedness in the solution results in a change of the total mass that is being *transported*.
> > > > >
> > > > > NubOT differs from this. Since we are learning a Monge map instead of a coupling, we always transport all points and our solution aims to always fit the marginals exactly (without a relaxation). However, to achieve this in an unbalanced setting, we introduce a weighting value for each point, which allows mass to be rescaled during transport (hence, the semi-coupling formulation). When using standard UBOT as a subroutine in our algorithm, we normalize the values (see lines 6 and 8 in Algorithm 1). This means that for both $\eta$ and $\zeta$, the sum of weights is approximately the number of points, i.e. a *total mass of 1* is being transported, distributed over the points. The resulting map we obtain is optimal in the following sense:
> > > > >
> > > > > 1. It minimizes the total re-weighted transport cost, i.e., $w(x) \||x-T(x)\||^2$,
> > > > > 2. .. while allowing a relative re-scaling of points via $w(x)$, designed in a way that $\frac{1}{n} \sum_\mathcal{X} w(x) = 1$,
> > > > > 3. .. while fulfilling the constraint $T_{\sharp} (\mathbf{w} \cdot \mu) = \nu$ without relaxation,
> > > > >
> > > > > where $w(x)$ is a weight value per point, in our case $\eta(x)  \cdot \zeta(T(x))^{-1})$
> > > > >
> > > > > Therefore, we get the robustness of the obtained weights w.r.t different parameter choices $\tau$.
> > > > >
> > > > > > **It might be helpful to add these elements one at a time.**
> > > > >
> > > > > Unfortunately, such type of ablation study is not possible here. All these steps are central components of NubOT, and not independent elements that can be exchanged. Using ICNNs to parameterize the Monge map is what allows us to robustly learn an OT map that approximates the target marginal, without a relaxation. To the best of our knowledge, there is no *standard balanced OT map* procedure that parameterizes the map. Instead, the solution based on ICNNs (Makkuva et al., 2020) is state-of-the-art on this task. This inherently includes learning the parameters in a mini-batch setting via SGD.
> > > > >
> > > > > A comparison between this balanced OT map via ICNNs and the re-weighted OT map via ICNNs is given by comparing our solution to CellOT, which uses the formulation of Makkuva et al. with improved initialization schemes.

---

> > > > > > ### Comment · Reviewer_CvpT · 2022-12-13
> > > > > > **Follow up**
> > > > > >
> > > > > > > In the standard UBOT algorithm,
> > > > > > τ
> > > > > >  controls the `unbalancedness' by specifying the penalty on the extent of marginal constraint violation. Allowing more unbalancedness in the solution results in a change of the total mass that is being transported.
> > > > > >
> > > > > > >NubOT differs from this.
> > > > > >
> > > > > > I'm not sure this is true, but perhaps I'm missing something. Aren't there a $\tau, \lambda$ (balancedness and entropy regularization) parameters in the `unbalanced_sinkhorn` calls in NubOT? I would think as this $\tau$ goes to infinity and $\lambda \to 0$ we should recover the standard balanced Monge problem, i.e. $w(x) \to \frac{1}{n}$ and equivalence with CellOT.
> > > > > >
> > > > > > > Unfortunately, such type of ablation study is not possible here.
> > > > > >
> > > > > > I agree with you for evaluations that extend to a validation set, but for the recovery of true unbalanced weights standard discrete OT can be used instead of the ICNN (i.e. in figures 1, 8, 9). This would help to validate the semi-couplings formulation independent of the ICNN optimization.

---

### Author Response · Authors · 2022-11-19
**Response to All Reviewers**

We thank the reviewers for their thorough and constructive feedback – it has undoubtedly strengthened the paper.

Our main paper has **significant** changes. Here are our **main highlights**:

### **Additional Baselines**

We have added additional baselines including the **closed-form of the entropy-regularized optimal transport problem on unbalanced Gaussians** by Janati et al., (2021) as well as **discrete OT**. We note that the latter is not operating in the out-of-sample setting, but serves as an additional reference to compare NubOT’s performance.

### **Extended Analysis of NubOT**

We extended and improved the comparison of NubOT to ubOT GAN on the synthetic data setups for which the ground-truth scaling factors are known (see Table 1 and Figure 2 in the updated manuscript).

We further evaluate its relationship to the solution returned by the generalized Sinkhorn algorithm by Chizat et al., (2018), which also serves as a subroutine within the NubOT algorithm. To that end, we conduct a **hyperparameter screen** of parameters (see §B1.1 and Fig. 7) of the generalized Sinhorn algorithm and demonstrate that **NubOT is more robust to different hyperparameter choices** than recovering the weights from the generalized Sinkhorn algorithm by Chizat et al., (2018) directly. For this, we further evaluate the sensitivity of Chizat et al., (2018) on different toy settings with known ground-truth scaling factors (see §B1.2 and Fig. 8).
We further assess NubOT’s dependence on the batch size.

Besides MMD, we have added the entropy-regularized Wasserstein metric as an **additional evaluation metric**.

### **Remark on the Balancedness of the Problem between Proxy Measures**

Several reviewers asked why the proxy measures $\tilde{\mu}$ and $\tilde{\nu}$ are guaranteed to be balanced. During computation, the measures are balanced (i.e., normalized) by construction, but this is trivial to achieve by just plain normalization. Indeed, lines 6 and 8 in Algorithm 1 show that $\sum_i e_i =n$ and $\sum_j z_j =m$, and dividing by $n$ and $m$ in line 9 respectively yields total mass 1 (we chose to write $\mathbf{e}$ and $\mathbf{z}$ in this way to have the loss function in Line 9 be more familiar to readers, but we are more than happy to simplify it if the reviewers think it will be clearer. Regardless, what really matters here is that the OT *problem we solve between these measures* is balanced. This is a subtle but important distinction that was not clearly articulated in the paper, and which we have now clarified. What this means is that since we assume all pointwise mass rescaling (e.g., cell birth and death in our application) has been accounted for when rescaling $\mu \mapsto \tilde{\mu}$ and $\nu \mapsto \tilde{\nu}$, we can then model the problem between $\tilde{\mu}$ and $\tilde{\nu}$ as a balanced OT problem. That is, the marginal constraints can now be imposed (i.e, no mass can be created or destroyed anymore), which justifies seeking a transport *map* between them. In summary, the key idea behind our approach is to learn a modification of our original measures that allows for a balanced OT problem to be solved between them.

### **Updated Manuscript**

Please have a look at the changes we added to the manuscript, in particular, in § 4, A, B, and Figures 2, 3, 7, 8, 9, 10.

---

### Decision · Program_Chairs · 2023-01-20

**Decision:**

Reject

**Justification For Why Not Higher Score:**

Essentially, this work is a new member of the neural network-assisted OT algorithms for aligning unpaired samples in an unbalanced setting. In my opinion, it is a combination of existing techniques (e.g., ICNN, semi-coupling, and unbalanced Sinkhorn), and the main contribution is combining existing techniques to improve the performance of a significant application. Besides the lack of novelty, my main concerns include: (1) as some reviewers mentioned, the complexity and the scalability of the method may be questionable because of the usage of the unbalanced Sinkhorn --- when the batch size is large, some numerical issues will happen according to my experience; (2) in the comparison on MMD, the performance of the proposed method is just slightly better than CellOT and GaussApprox. To my knowledge, GaussApprox is much more computationally-efficient than the proposed method when the dimension of the feature is not very large. After reading the authors' reply, my concerns above are not solved.

The reviewers also proposed some comments about the novelty and the complexity of the proposed method. Although the authors tried to resolve these concerns, some reviewers are not very satisfied with the replies. Two reviewers raised their scores from 5 to 6, but one had low confidence.

**Justification For Why Not Lower Score:**

N/A

**Metareview: Summary, Strengths And Weaknesses:**

In this submission, the authors proposed a new member of the neural network-assisted OT algorithms for aligning unpaired samples in an unbalanced setting. Experiments on single-cell data are designed to demonstrate the usefulness of the proposed method.

Strengths:
(1) The idea is clear, and the derivation is easy to follow. Parametrizing OT plans via neural networks is an attractive topic in the community.
(2) The application considered by this submission, i.e., single-cell data analysis, is important.

Weaknesses:
(1) The novelty of the proposed method is limited, as the reviewers mentioned.
(2) The numerical stability and the scalability of the proposed method are not verified, especially when the sample size is large and the feature dimension is high.